# Comparative Study of Time Series Analysis Algorithms Suitable for Short-Term Forecasting in Implementing Demand Response Based on AMI

**DOI:** 10.3390/s24227205

**Published:** 2024-11-11

**Authors:** Myung-Joo Park, Hyo-Sik Yang

**Affiliations:** Department of Computer Science and Engineering, Sejong University, 209, Neungdong-ro, Gwangjin-gu, Seoul 05006, Republic of Korea; taesachi@gabia.com

**Keywords:** smart grid, AMI, demand response, ARIMA, SARIMA, LSTM, SVM, short-term forecasting

## Abstract

This paper compares four time series forecasting algorithms—ARIMA, SARIMA, LSTM, and SVM—suitable for short-term load forecasting using Advanced Metering Infrastructure (AMI) data. The primary focus is on evaluating the applicability and performance of these forecasting models in predicting electricity consumption patterns, which is a critical component for implementing effective demand response (DR) strategies. The study provides a comprehensive analysis of the predictive accuracy, computational efficiency, and scalability of each algorithm using a dataset of real-time electricity consumption collected from AMI systems over a designated period. Through extensive experiments, we demonstrate that each algorithm has distinct strengths and weaknesses depending on the characteristics of the dataset. Specifically, SVM exhibited superior performance in handling nonlinear patterns and high volatility, while SARIMA effectively captured seasonal trends. LSTM showed potential in modeling complex temporal dependencies but was sensitive to hyperparameter settings and required a substantial amount of training data. This research offers practical guidelines for selecting the optimal forecasting model based on data characteristics and application needs, contributing to the development of more efficient and dynamic energy management strategies. The findings highlight the importance of integrating advanced forecasting techniques into smart grid systems to enhance the reliability and responsiveness of DR programs. This study lays a solid foundation for future research on integrating these forecasting models into real-world AMI applications to support effective demand response and grid stability.

## 1. Introduction

Demand response (DR) is an essential component of smart grid systems, aimed at maintaining a dynamic balance between electricity demand and supply while enhancing overall grid efficiency and stability [1]. As global energy demand continues to rise and renewable energy sources become more prominent, implementing effective DR strategies has become increasingly critical for optimizing energy distribution and minimizing grid stress during peak periods.

DR strategies are typically divided into two main categories: Price-based demand response and incentive-based demand response. Price-based DR utilizes variable electricity pricing schemes, such as real-time pricing (RTP) and time-of-use (TOU) pricing, to influence consumer consumption patterns by adjusting prices based on demand fluctuations [2]. Incentive-based DR, on the other hand, provides direct economic incentives for consumers to modify their consumption behaviors during peak load times or critical periods, examples of which include demand response resources (DRR) and capacity market payments [3,4].

One of the key technologies enabling effective DR implementation is the Advanced Metering Infrastructure (AMI), a foundational component of modern smart grids. AMI systems facilitate two-way communication between consumers and utility providers, enabling the collection and real-time analysis of granular electricity consumption data. This capability is essential for implementing dynamic DR programs and for ensuring the effectiveness of both price-based and incentive-based DR strategies [5]. However, due to the distributed nature of AMI systems and the high variability of consumption patterns, accurate short-term load forecasting poses significant challenges, particularly when attempting to predict nonlinear and volatile trends in energy usage.

To address these challenges, this study proposes a comparative analysis of four widely used time series forecasting models—ARIMA, SARIMA, LSTM, and SVM—specifically tailored for AMI-based short-term load forecasting. This research contributes to the field by providing an in-depth evaluation of the performance, computational complexity, and scalability of each model under varying data conditions. A central focus of the study is to identify the optimal forecasting model for supporting distributed DR strategies, taking into account the unique characteristics of AMI systems and the need for real-time prediction capabilities.

The primary contributions of this research are summarized as follows:Comprehensive model evaluation: a detailed evaluation of the forecasting performance of ARIMA, SARIMA, LSTM, and SVM models for short-term load forecasting in AMI-based DR scenarios.dentification of model suitability: insights into the suitability of each model for different DR strategies, highlighting the trade-offs between prediction accuracy, computational requirements, and scalability.

In summary, this study provides a comprehensive analysis of time series forecasting models for short-term load forecasting in AMI systems, offering valuable guidance for utility companies and researchers in selecting the most suitable predictive models for enhancing energy management in smart grid environments.

## 2. Related Work

Advanced Metering Infrastructure (AMI) represents a pivotal component of the smart grid. Its primary function encompasses tracking electricity consumption at client nodes and transmitting this information to utility centers. AMI undertakes roles in information collection, file storage, file transmission, bandwidth allocation, and billing issuance [6]. Additionally, Wang et al. [7] highlight the accuracy and reliability as significant advantages of demand response implementations in AMI systems:Accuracy: AMI-based demand response systems offer high precision in load control, contributing to efficient energy management and grid stability.Reliability: the centralized architecture and control strategies of AMI facilitate stable and trustworthy demand response operations, ensuring consistent performance.

However, smart meters within AMI are resource-constrained devices, characterized by limited computational and storage capabilities [8]. Moreover, a major challenge for AMI systems is their sustainability under unexpected usage patterns, low bandwidth, and significant informational performance degradation, with low-power-centric data technologies at the root of these issues [6]. Therefore, research in AMI functionalities must consider the constraints imposed by low-power-centric data technologies, particularly the limitations in computing and storage capabilities, which warrant serious consideration. Thus, determining the appropriate duration for data collection and analysis applied to demand response forecasting becomes crucial. While there is no standardized period for data collection and analysis for forecasting, various studies suggest differing durations: for short-term load forecasting (STLF), the period ranges from less than a day to a week [9,10,11]; medium-term load forecasting (MTLF) spans two weeks to three years [9,10,11]; and long-term load forecasting (LTLF) encompasses three years to fifty years [9,11]. According to research by Hong et al. and Yunsun Kim et al., while no standard period exists, it is common to set STLF to less than a week, MTLF to one week to a month, and LTLF to more than a year [12,13]. Consequently, research into demand response in AMI must prioritize the selection of algorithms suitable for short-term forecasting, considering the limitations of low-power-based technologies and the potential applicability of each algorithm to short-term forecasting and demand response.

ARIMA is a traditional time series forecasting model, widely used for non-stationary data. It combines autoregressive (AR) terms, integrated (I) differencing, and moving average (MA) terms to predict future values [14,15]. SARIMA extends ARIMA by incorporating seasonal components. It is used for datasets with clear seasonal patterns [16,17]. Recent research papers reveal that ARIMA and SARIMA models are effectively utilized in short-term forecasting, particularly in fields such as electric load forecasting and mobile network traffic predictions. These studies underscore the models’ efficacy in accurately predicting short-term variations across various sectors. For instance, research conducted by Kochetkova et al. employed SARIMA and other models for short-term forecasting of enterprise network traffic, demonstrating potential applicability in the electricity sector [18]. Minnaar et al. focused their research on using the SARIMA model for forecasting in electricity distribution, procurement, and sales, emphasizing the importance of accurate load forecasting for utility companies, especially in the context of increasing roles of renewable energy sources [19]. Moreover, studies like the one by Kien et al., which addressed SARIMA model usage for electric demand forecasting in Hanoi, showcase its effectiveness in predicting electricity demand in specific regions [20]. These examples demonstrate that ARIMA and SARIMA models are widely recognized and used across various sectors, making them highly relevant to research themes focusing on AMI and demand response.

Long Short-Term Memory (LSTM) models are deep learning models capable of effectively handling long-term dependencies in complex time series data [17]. Developed to overcome the long-term dependency issues inherent in traditional Recurrent Neural Networks (RNNs), LSTMs have provided effective forecasting results in diverse areas, including electric load forecasting, stock market analysis, and weather condition predictions. A study by Al-Musaylh et al. that applied LSTM for electric demand forecasting illustrates the model’s capability in handling complex time series data [21]. Recent studies have explored LSTM’s approach to short-term forecasting across various fields. For example, research by Vatsa et al. employed LSTM for short-term forecasting of polarization current for transformer insulation assessment, extending LSTM architecture to capture long-term dependencies and complex temporal patterns [22].

Support Vector Machine (SVM) is a powerful machine learning algorithm effective for handling nonlinear datasets [23]. Primarily known for classification problems, SVM is a robust machine learning technique that can also be applied to regression problems. It operates by mapping data features to a higher-dimensional space to find the optimal decision boundary. This characteristic makes it particularly valuable for short-term forecasting of nonlinear and complex data patterns. For instance, a study by Nepal et al. employing SVM for electric load forecasting demonstrated the model’s effectiveness in handling complex data patterns and providing accurate predictions [24].

However, existing studies tend to focus on implementing and examining a single algorithm. Hence, this paper aims to compare traditional time series analysis methods, such as ARIMA and SARIMA, with algorithms like LSTM and SVM to enhance the efficiency of short-term forecasting in AMI systems. This comparison of diverse algorithms will play a crucial role in improving data processing and forecasting accuracy in AMI systems, ultimately contributing to better decision-making in energy management and planning.

## 3. Research Methodology

This section outlines the unique approach used in this study to compare various forecasting algorithms for AMI-based short-term demand response. While ARIMA, SARIMA, LSTM, and SVM are well-known models, our study uniquely applies them within the context of distributed AMI systems to determine which model best captures the nonlinear, volatile nature of electricity consumption patterns. The results from this comparison provide important insights into the suitability of each model for real-world applications in dynamic energy management systems.

### 3.1. Application of Algorithms

This section meticulously delineates the implementation methodology for each of the selected time series forecasting algorithms.

#### 3.1.1. ARIMA (Auto Regressive Integrated Moving Average)

The ARIMA model is employed to transform non-stationary time series data into a stationary form for future value predictions. The implementation of the ARIMA model encompasses the following steps:1.Identify and preprocess time series data to remove trends and seasonality.(a)Removing trends: Linear or nonlinear trends are eliminated from the data through differencing. The first differencing is calculated as follows:
(1)$$y_{t}^{\prime }=y_{t}-y_{t-1}$$
where $y_{t}$ is the observation at time *t*, and $y_{t}^{\prime }$ is the differenced time series.(b)Removing seasonality: seasonal differencing is applied to eliminate seasonal variations in the data, calculated over a period *S*:
(2)$$y_{t}^{\prime \prime }=y_{t}-y_{t-S}$$
where *S* represents the seasonal period.(c)Checking stationarity: After differencing, stationarity of the time series is tested using the ADF test. If the p-value is below 0.05, the series can be considered stationary.(d)Applying transformations: log transformation or Box-Cox transformation is applied to stabilize the variance of the data.(e)Analyzing ACF and PACF plots: the parameters *p*, *d*, and *q* of the ARIMA model are determined by analyzing the ACF and PACF plots.2.Analyze ACF and PACF plots to determine appropriate AR(p) and MA(q) parameters.(a)Analyzing ACF plot: The ACF plot visualizes the correlation between observations in the time series data, showing the total correlation between lagged observations, and is used to estimate the order of the MA model (q). The cut-off point in the ACF plot after which correlations rapidly decrease can be chosen as the value for MA(q).(b)Analyzing PACF plot: The PACF plot shows the correlation of individual lag values excluding the influence of other lag values, primarily used to determine the order of the AR model (p). The cut-off point in the PACF plot after which correlations rapidly decrease can be chosen as the value for AR(p).(c)Parameter estimation: Based on ACF and PACF plots, the values of AR(p) and MA(q) best fitting the data are determined experimentally. Various combinations of p and q are tried, using information criteria (e.g., AIC, BIC) to select the optimal model.3.Apply differencing (d) to make the data stationary and fit the ARIMA model to the data:(a)Differencing: differencing in the ARIMA model is used to transform non-stationary time series data into a stationary form.(b)Model fitting: Fit the ARIMA model to the differenced data. The ARIMA(p, d, q) model is defined as follows:
(3)$$\left(1-\sum_{i=1}^{p}\phi_{i}L^{i}\right){\left(1-L\right)}^{d}y_{t}=\left(1+\sum_{i=1}^{q}\theta_{i}L^{i}\right)\varepsilon_{t}$$(c)Model estimation: use optimization techniques to estimate the values of the model parameters ϕ and θ, evaluating the model’s fit and adjusting parameters as necessary.4.Validate the model’s stationarity using statistical tests like the ADF test.5.Use the final model to predict future values and evaluate the model’s performance.(a)Predicting future values: Use the fitted ARIMA model to predict future values. The prediction is conducted using the model’s mathematical expression.(b)Evaluating model performance: Analyze the difference between predicted and actual values to evaluate the model’s performance. Common metrics include RMSE or MAE for performance evaluation.(c)Model Validation: Validate the model’s fit based on prediction performance and adjust the model as necessary. Information criteria such as AIC or BIC can be used for comparison with other models.Through these steps, the predictive capability of the ARIMA model is assessed, and its applicability to real-time series data is confirmed.

According to reference [17], the application of the ARIMA model to order demand forecasting demonstrates the model’s applicability.

#### 3.1.2. SARIMA (Seasonal ARIMA)

The SARIMA model extends the ARIMA model by incorporating seasonality, making it suitable for forecasting data with seasonal variations. The implementation of the SARIMA model includes the following steps:1.Identify and preprocess time series data, including seasonal elements.(a)Identify seasonal elements and preprocess the data accordingly. This involves capturing clear seasonal patterns in the time series data and normalizing them through differencing.(b)Apply seasonal differencing to normalize the data, defined as follows:
(4)$$y_{t}^{\prime }=y_{t}-y_{t-S},$$
where *S* is the seasonal period.2.Determine the seasonal parameters (P, D, Q) of the SARIMA model based on the seasonal cycle (S):Decide on the seasonal parameters of the SARIMA model based on the seasonal cycle *S*, used to model periodic fluctuations in seasonal patterns.Analyze seasonal ACF and PACF plots to determine seasonal AR and MA parameters.3.Fit the model to the data and further adjust non-seasonal parameters (p, d, q) using ACF and PACF plots.4.Evaluate the model’s fit through residual analysis and adjust the model if necessary:(a)Calculate residuals:
(5)$$e_{t}=y_{t}-{\hat{y}}_{t},$$
where $e_{t}$ are the residuals, $y_{t}$ the observations, and ${\hat{y}}_{t}$ the predictions.(b)Assess statistical properties of residuals:Verify that the mean of the residuals is close to zero.Check for constant variance in the residuals (homoscedasticity).Examine residuals for autocorrelation using ACF and PACF.(c)Perform a normality test:Test whether the distribution of residuals follows a normal distribution (e.g., Shapiro–Wilk test).Residual analysis evaluates the model’s fit and allows for adjustments to improve prediction accuracy.5.Generate forecasts considering seasonality using the final model. Predictions with the SARIMA model are made based on current and past data points and model parameters, estimating values for future points. The prediction formula is as follows:
(6)$${\hat{y}}_{t+h}=\mu +\sum_{i=1}^{P}\Phi_{i}y_{t+h-iS}+\sum_{i=1}^{p}\phi_{i}y_{t+h-i}-\sum_{i=1}^{Q}\Theta_{i}\varepsilon_{t+h-iS}-\sum_{i=1}^{q}\theta_{i}\varepsilon_{t+h-i}$$
where the following is true:${\hat{y}}_{t+h}$ is the forecast value at time t+h.μ is the mean of the model.$\Phi_{i}$, $\phi_{i}$, $\Theta_{i}$, $\theta_{i}$ are the seasonal and non-seasonal AR and MA parameters, respectively.$y_{t+h-iS}$ and $y_{t+h-i}$ are past observations, and $\varepsilon_{t+h-iS}$ and $\varepsilon_{t+h-i}$ are past error terms.Using this formula, the SARIMA model predicts future values considering past data and seasonal patterns.

According to reference [25], the SARIMA model was highly effective in capturing seasonal patterns in time series data.

#### 3.1.3. LSTM (Long Short-Term Memory)

LSTM is a deep learning-based time series prediction model capable of learning long-term dependencies in data. The implementation of the LSTM model includes the following steps:1.Normalize the time series data and prepare them by dividing sequences into windows for feeding into the network.(a)Data normalization: The process of normalizing data to enhance the efficiency of model training. Min–max normalization is the most common method.
(7)$$x_{\mathrm{normalized}}=\frac{x-\min(x)}{\max(x)-\min(x)}$$
where *x* is the original data value and min(x) and max(x) are the minimum and maximum values of the dataset, respectively.(b)Sequence window splitting: Time series data are divided into continuous sequences to be learnable by the network. Each sequence consists of a specific length of windows.
(8)$$X_{t}=\left[x_{t-n},\ldots ,x_{t-1}\right],Y_{t}=x_{t},$$
where $X_{t}$ is the input sequence, $Y_{t}$ is the value to be predicted at the next time point, and *n* is the window size.2.Define the LSTM architecture, determining the number of layers and the number of neurons in each layer.(a)Number of layers: The number of layers in the LSTM network determines the model’s depth, typically adjusted based on the complexity of the problem. Using more layers allows the model to learn more complex patterns but may increase the risk of overfitting.(b)Number of neurons per layer: The number of neurons in each LSTM layer determines the capacity of that layer. More neurons allow for storing more information but increase computational costs.(c)Architecture example:     model = Sequential()     model.add(LSTM(50,     return_sequences=True,     input_shape=(n_input, n_features)))     model.add(LSTM(50,     return_sequences=False))     model.add(Dense(1))     model.compile(optimizer=’adam’,     loss=’mse’)In this example, two LSTM layers are used with 50 neurons each. ‘return_sequences = True‘ indicates that the output of the first layer is a sequence. The number of layers and neurons per layer should be optimized through experimentation to maintain a balance between performance and efficiency of the model.3.Apply techniques to prevent overfitting using the prepared data.(a)Overfitting: Refers to a situation where the determined weights produce different results for new training data. As illustrated in Figure 1, overfitting tends to occur more frequently when there is a smaller amount of data available for training [26].(b)Measures to prevent overfitting include the following methods:Cross-validation: A method that uses a portion of the training data as validation data [25]. However, for time series data, the order of the data over time must be considered, requiring the use of nested cross-validation [27] in such cases.Dropout: In a fully connected neural network as shown in Figure 2, using a dropout layer as depicted in Figure 3 removes parts of the neural network for training purposes [28].4.Validate the trained model’s performance, measuring prediction accuracy using evaluation metrics.5.Perform predictions on future data using the validated model.

Reference [25] presents a case of applying LSTM for time series forecasting, demonstrating LSTM’s ability to effectively learn and predict complex data patterns.

#### 3.1.4. SVM (Support Vector Machine)

SVM is a powerful machine learning model used for time series forecasting, particularly effective in handling nonlinear and complex data patterns. It employs a risk function based on the principle of structural risk minimization to balance empirical error and regularization.

The implementation of SVM in short-term forecasting research based on AMI involves the following steps:1.Data preparation: preprocess and normalize AMI data to make them suitable for SVM.2.Model development: configure SVM using an appropriate kernel function to capture the complexity of the data.3.Training and testing: train the SVM model using historical AMI data and test its performance on unseen data.4.Performance evaluation: assess the model’s accuracy, speed, and efficiency using metrics appropriate for time series analysis.

Through this approach, SVM can be a useful tool in AMI systems for handling complex and nonlinear data patterns.

SVM is a powerful machine learning algorithm that defines a decision boundary based on the characteristics of the dataset, finding a hyperplane with the maximum margin to classify the given data. The basic form of SVM applies to linear classification problems, but it can be extended to nonlinear classification problems using the kernel trick.

The objective function of SVM is defined as follows:(9)$$\min_{w,b}\frac{1}{2}{∥w∥}^{2}+C\sum_{i=1}^{n}\xi_{i}$$

Here, w is the weight vector of the hyperplane, *b* is the bias, $\xi_{i}$ are slack variables, and *C* is the regularization parameter. This objective function aims to find a hyperplane that maximizes the margin and minimizes classification errors.

The kernel trick allows modeling complex nonlinear relationships by mapping the input space into a higher-dimensional feature space. The most commonly used kernel function is the Radial Basis Function (RBF):(10)$$K\left(x_{i},x_{j}\right)=\exp(-\gamma ∥x_{i}-x_{j}{∥}^{2})$$

Here, γ is the parameter of the RBF kernel, and $x_{i}$ and $x_{j}$ are feature vectors.

This research applies SVM to short-term electric demand forecasting, similar to methodologies proposed by Shi et al. [29] and Sun et al. [30]. These studies highlight the efficiency and accuracy of SVM, supporting the approach of this research.

## 4. Experiments and Results

This section presents the experimental methods, the data used, and the results obtained from evaluating the performance of the models. The models were assessed using standard forecasting error metrics: Mean Squared Error (MSE), Mean Absolute Error (MAE), and Root Mean Squared Error (RMSE). These metrics are widely used in forecasting studies and offer a reliable basis for comparing model accuracy. Tables 7–16 summarize the key results for each model across multiple datasets. A detailed analysis is also provided to explore the trade-offs between accuracy and computational complexity.

### 4.1. Experimental Methodology

The experiments in this research will be conducted as follows:1.Collection and trend analysis of time-specific AMI measurement data: collect data for training and analyze trends to understand the actual movements within the AMI measurement data.(a)Data collection for training.(b)Data collection for comparison.2.Applying collected data to each algorithm: the collected data are applied to each algorithm.3.Trend comparison: the trends from the results of applying each algorithm are compared with previously observed trends to analyze differences.4.Data comparison: directly compare the data collected for comparison with the results from each algorithm to analyze the differences.

Through the above processes, the performance of each algorithm will be indirectly assessed to determine which algorithm performs best. Additionally, considering the operation based on AMI, the experiments will be conducted in a virtual environment configured with a 1 Ghz 2 core CPU and 800 MB RAM.

### 4.2. Introduction to Experimental Data

KT, the National Information Society Agency (NIA), and the Ministry of Science and ICT of South Korea provide big data across various industrial sectors (https://www.bigdata-telecom.kr/; accessed on 28 January 2024). The big data used in this experiment are as follows:1.Data for comparison: time-specific AMI AI training data.(a)Data period: 1December 2021 to 1 January 2022.(b)Data size: 5,426,322,257 bytes.2.Data for training: time-specific AMI electricity usage.(a)Data period: 1 July 2021 to 1 December 2021.(b)Data size: 15,479,044,282 bytes.

The data collected for final comparison is named “Time-Specific AMI AI Training Data,” and the data collected for training is named “Time-Specific AMI Electricity Usage.”

Table 1 defines the “Time-Specific AMI AI Training Data” file, intending to verify the data and its trends based on the consumer number (CNSMR_NO), Measurement Date (MESURE_DE), and Measurement Time (MESURE_TM).

### 4.3. Introduction to the Experimental Method for Learning AMI Time-Specific Power Usage Data

The collected “AMI Time-Specific Power Usage Data,” like the previously collected “AMI Time-Specific AI Learning Data” for comparison, is provided in csv format, representing data measured in apartments equipped with actual AMI devices, organized by time of day. The data from 1 July 2021 to 1 December 2021 will be used for training each algorithm, and the results will be saved in csv files in the same format as the “AMI Time-Specific AI Learning Data” for direct comparison in the previous subsection, “Collection and Trend Analysis of Time-Specific AMI Measurement Data.” To facilitate this, data files from 1 July 2021 to 1 December 2021 were merged into one file, resulting in a csv file with a total size of 15,479,044,282 bytes and containing 316,712,453 lines. From this file, data for the 10 consumer numbers (CNSMR_NO) sampled in Figure 4 will be extracted. The definition of this recreated csv format can be found in Table 2.

The code to be implemented for the experiment and the hardware specifications on which the experiment will be conducted are as follows:1.Language: Python 3.11.72.Used libraries are as follows:(a)pandas.(b)pmdarima.(c)numpy.(d)tqdm.(e)datetime.(f)sys.(g)sklearn.(h)tensorflow.3.Virtual environment hardware specifications (virtual machine) are as follows:(a)CPU: 1.0 GHz 2 core.(b)RAM: 800 MB.

### 4.4. Data Description

The data used in this study are derived from Advanced Metering Infrastructure (AMI) systems, and they include a variety of fields that capture different aspects of electricity usage. Each field serves a specific purpose in the data preprocessing and feature engineering steps, ultimately influencing the prediction accuracy of the algorithms. Below, we briefly describe the role of each key field in Table 1 and Table 2:-CNSMR_NO (Consumer number): A unique identifier for each household used to track electricity consumption patterns. This field helps group the data for individual time series analysis, enabling models to capture household-specific trends.-LEGALDONG_CD (Legal district code): This code indicates the geographical location of the household. It is used to analyze the impact of regional differences in energy usage and is particularly useful for capturing spatial variability in consumption patterns.-MESURE_DE (Measurement Date) and MESURE_TM (Measurement Time): These fields record the exact date and time of each measurement, which is critical for capturing daily and seasonal patterns. Temporal fields such as these are essential for time-series models to detect and predict seasonal variations and trends.-TOC_NO (Type of consumer cumber): An identifier indicating the type of consumer (e.g., residential, commercial) to classify consumption patterns according to consumer categories. It helps in segmenting data for category-specific analysis.-HLDY_AT (Holiday indicator): This binary field indicates whether the measurement was taken on a holiday, which is crucial for understanding variations in energy consumption on non-working days. It is used as an external regressor in models like SARIMA.-SG_PWRER_USE_AM (Power usage amount): The primary target variable representing the amount of power consumed. This field is the key variable for all forecasting models and serves as the response variable in regression-based algorithms.

### 4.5. ARIMA Algorithm Experiment and Results

Algorithm 1 presents the actual implementation, encompassing the majority of the steps involved in the implementation process of the proposed ARIMA model.
**Algorithm 1** Forecasting and Saving Power Usage Data Using ARIMA1:**function** PreprocessChunk(chunk)2:    Combine ’MESURE_YEAR’, ’MESURE_MT’, ’MESURE_DTTM’ into ’Date’3:    **return** chunk4:**function** LoadAndPreprocessData (filepath, chunksize)5:    Load data in chunks from filepath using specified columns6:    Apply PreprocessChunk to each chunk7:    Concatenate all processed chunks into a single DataFrame8:    **return** the concatenated DataFrame9:**function** AutoSelectOrderAndPredict(df, n_periods)10:    Initialize an empty list for results11:    **for** each group in df grouped by ’LEGALDONG_CD’ and ’CNSMR_NO’ **do**12:        Extract ’SGPowerUsage’ series from the group13:        Fit auto ARIMA model and predict for n_periods14:        Append forecast results to the results list15:    **return** list of forecast results16:**function** SaveForecast(results, output_file, start_of_next_month, n_periods)17:    Generate date and hour range for the next month starting from start_of_next_month18:    Convert results to a DataFrame with proper date and hour columns19:    Save the DataFrame to output_file20:**procedure** Main(input_csv_file, output_csv_file)21:    df← Load and preprocess data from input_csv_file22:    Calculate the start of next month and the number of periods (n_periods)23:    results← Perform ARIMA forecasting for each group in df24:    Save forecast results to output_csv_file

Table 3 summarizes the slope and intercept of the calculated trend lines for each consumer number (CNSMR_NO).

### 4.6. SARIMA Algorithm Experiment and Results

Algorithm 2 presents the actual implementation, encompassing the majority of the steps involved in the implementation process of the proposed SARIMA model.
**Algorithm 2** Forecasting Power Usage with Seasonality and Saving the Results1:**function** PreprocessChunk(chunk)2:    Combine ’MESURE_YEAR’, ’MESURE_MT’, ’MESURE_DTTM’, ’Hour’ into ’DateTime’3:    Create a dummy variable ’IsHoliday’ from ’HLDY_AT’4:    **return** chunk5:**function** LoadAndPreprocessData(filepath, chunksize)6:    Load data in chunks from filepath using specified columns7:    Apply PreprocessChunk to each chunk8:    Concatenate all processed chunks into a single DataFrame9:    **return** the concatenated DataFrame10:**function** AutoSelectOrderAndPredict(df, n_periods)11:    Initialize an empty list for results12:    **for** each group in df grouped by ’LEGALDONG_CD’ and ’CNSMR_NO’ **do**13:        Prepare the time series and holiday series sorted by ’DateTime’14:        Fit SARIMA model with holiday information as an exogenous variable15:        Forecast n_periods ahead with the last week’s holiday data16:        Append forecast results and dates to the results list17:    **return** list of forecast results18:**function** SaveForecast(results, output_file)19:    Convert results to a DataFrame with date and hour separated20:    Save the DataFrame to output_file21:**procedure** Main(input_csv_file, output_csv_file)22:    df← Load and preprocess data from input_csv_file23:    Calculate the start of next month and determine n_periods for forecasting24:    results← Perform SARIMA forecasting for each group in df25:    Save forecast results to output_csv_file

Table 4 summarizes the slope and intercept of the calculated trend lines for each consumer number (CNSMR_NO).

### 4.7. LSTM Algorithm Experiment and Results

Algorithm 3 presents the actual implementation, encompassing the majority of the steps involved in the implementation process of the proposed LSTM model.
**Algorithm 3** Forecasting Power Usage with LSTM Model for Each Consumer Group1:**function** PreprocessData(df)2:    Normalize ’SGPowerUsage’ using MinMaxScaler3:    **return** df, scaler4:**function** CreateSequences(df, n_steps)5:    Initialize empty arrays for *X* (features) and *y* (labels)6:    **for** each *i* from n_steps to len(df) **do**7:        Append scaled usage values from i−n_steps to *i* to *X*8:        Append scaled usage value at *i* to *y*9:    **return** *X*, *y*10:**function** BuildLSTMModel(input_shape)11:    Create Sequential model with LSTM and Dense layers (units=50)12:    Compile model with ’adam’ optimizer and mean squared error loss13:    **return** model14:**function** PredictForEachGroup(group, n_steps, model, scaler)15:    Initialize list for predictions16:    Prepare input sequence for the next month’s first prediction17:    **for** each hour in next month **do**18:        Predict next hour’s usage and inverse transform the scaled value19:        Append prediction to the list20:        Update input sequence with the new prediction21:    **return** predictions22:**procedure** Main(input_csv_file, output_csv_file, n_steps)23:    Load and preprocess data from input_csv_file24:    Group data by ’LEGALDONG_CD’ and ’CNSMR_NO’25:    Initialize list for all predictions26:    **for** each group **do**27:        Sort group by date and reset index28:        **if** group size ≥n_steps **then**29:           *X*, *y*←CreateSequences(group, n_steps)30:           model←BuildLSTMModel((n_steps, 1))31:           Train model with early stopping32:           predictions←PredictForEachGroup(group, n_steps, model, scaler)33:           Append predictions to the all predictions list34:    Save all predictions to output_csv_file

Table 5 summarizes the slope and intercept of the calculated trend lines for each consumer number (CNSMR_NO).

### 4.8. SVM (Support Vector Machine) Algorithm Experiment and Results

Algorithm 4 presents the actual implementation, encompassing the majority of the steps involved in the implementation process of the proposed SVM model.
**Algorithm 4** Predict Future Power Usage Using SVR Model1:**function** PreprocessData(filepath)2:    Load data from filepath into DataFrame df3:    Convert date columns to ’Date’ in yyyymmdd format4:    Ensure ’Hour’ is integer5:    **return** df6:**function** GenerateFutureDates(last_date_str, n_days)7:    Calculate start of next month from last_date_str8:    Generate hourly dates for n_days into the future9:    **return** list of future dates10:**function** PrepareFutureDataset(df, future_dates)11:    Create empty DataFrame for future data12:    **for** each date in future_dates **do**13:        Create temporary DataFrame for the date14:        Append to future data DataFrame15:    **return** DataFrame for future predictions16:**function** TrainAndPredict(df)17:    Initialize list for predictions18:    **for** each group in df by ’CNSMR_NO’ and ’LEGALDONG_CD’ **do**19:        Separate features and target20:        Scale features using StandardScaler21:        Split data into training and test sets22:        Train SVR model on training set23:        Prepare dataset for future predictions24:        Scale future features and predict future power usage25:        Append predictions to list26:    Combine all predictions into a DataFrame27:    **return** DataFrame with predictions28:**procedure** Main(input_csv_file, output_csv_file)29:    df← PreprocessData(input_csv_file)30:    predictions_df← TrainAndPredict(df)31:    Save predictions_df to output_csv_file

Table 6 summarizes the slope and intercept of the calculated trend lines for each consumer number (CNSMR_NO).

### 4.9. Experimental Results and Comparison

From Figure 5, Figure 6, Figure 7, Figure 8, Figure 9, Figure 10, Figure 11, Figure 12, Figure 13 and Figure 14, graphs feature a red zero line. This line indicates that the closer the predicted data, represented in blue, are to zero, the closer they are to the actual data. Furthermore, the predicted results for each consumer number (CNSMR_NO) will be quantitatively evaluated using three evaluation methods: MSE, MAE, and RMSE.

1.Mean Squared Error (MSE)(a)MSE: the difference between the actual and predicted values are squared, averaged, and then calculated.
(11)$$\mathrm{MSE}=\frac{1}{n}\sum_{i=1}^{n}{\left(Y_{i}-{\hat{Y}}_{i}\right)}^{2}$$(b)Here, $Y_{i}$ represents the actual value, ${\hat{Y}}_{i}$ the predicted value, and *n* the number of samples. Since MSE squares the errors, it gives more weight to larger errors, making it a useful tool for emphasizing model performance in situations where larger errors are disadvantageous. Therefore, it has the characteristic of being more sensitive to larger errors [31].2.Mean Absolute Error (MAE)(a)MAE: defined as the average of the absolute differences between the predicted values and the actual values.
(12)$$\mathrm{MAE}=\frac{1}{n}\sum_{i=1}^{n}\left|Y_{i}-{\hat{Y}}_{i}\right|$$(b)Unlike MSE, MAE evaluates model errors linearly, making it less sensitive to larger errors. Therefore, this characteristic makes it a metric that can be used when all errors need to be treated uniformly. It differentiates itself from MSE by treating all errors equally [31].3.Root Mean Squared Error (RMSE)(a)By taking the square root of MSE, the utility of MSE can be further extended, thereby restoring the error metric to the same scale as the predicted data.
(13)$$\mathrm{RMSE}=\sqrt{\frac{1}{n}\sum_{i=1}^{n}{\left(Y_{i}-{\hat{Y}}_{i}\right)}^{2}}$$(b)RMSE combines the sensitivity of MSE to large errors with the advantage of being in the same units as the response variable, making it an essential metric for comparing and evaluating various regression models on a consistent scale [31].

In the domain of predictive modeling and data analysis, the Mean Squared Error (MSE), Mean Absolute Error (MAE), and Root Mean Squared Error (RMSE) serve as fundamental metrics for evaluating the performance and accuracy of regression models. Each metric offers a distinctive approach to quantifying and analyzing the discrepancies between the model’s predicted values and the actual observed data, thereby enabling a robust understanding of the model’s effectiveness in relation to empirical data. Algorithm 5 represents the implementation code for each analysis method.
**Algorithm 5** Calculate Error Metrics per CNSMR_NO1:**procedure** LoadData(predicted_path, actual_path)2:    predicted_data← Load CSV from predicted_path without headers3:    actual_data← Load CSV from actual_path without headers4:    Set column names for both datasets5:    **return** predicted_data, actual_data6:**procedure** MergeData(predicted_data, actual_data)7:    Merge on ’CNSMR_NO’, ’Date’, ’Hour’ with suffixes8:    **return** merged_data9:**function** CalculateMetrics(group)10:    error← Calculate difference between predictions and actuals11:    mse← Calculate mean squared error12:    mae← Calculate mean absolute error13:    rmse← Calculate root mean squared error14:    **return** mse, mae, rmse15:**procedure** Main(predicted_csv, actual_csv)16:    predicted_data, actual_data ← LoadData(predicted_csv, actual_csv)17:    merged_data← MergeData(predicted_data, actual_data)18:    Initialize error_metrics as an empty dictionary19:    **for** each consumer_no in merged_data grouped by ’CNSMR_NO’ **do**20:        mse, mae, rmse← CalculateMetrics(group)21:        Store mse, mae, rmse in error_metrics with key consumer_no22:    Print error_metrics

Figures illustrating the differences between predicted and actual data and Algorithm 5 employing each algorithm’s predicted and actual data to assess accuracy using the MSE, MAE, and RMSE methods are followed by tables summarizing the evaluation metrics. The overall analysis based on these evaluation metrics yields the following results.

1.Consumer number (CNSMR_NO): DJ0200309001501.(a)Figure 5: graphs illustrating the differences between algorithm-specific predicted and actual data.
Figure 5Differences in SGPowerUsage values between predictions and actual data in DJ0200309001501.
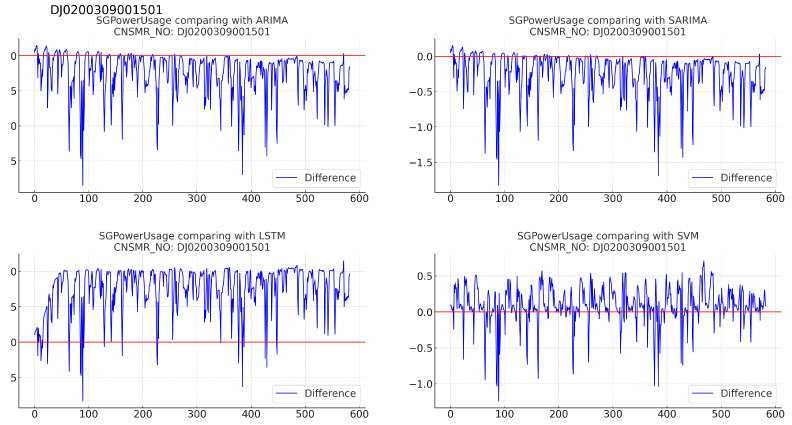
(b)Table 7: accuracy evaluation for the predicted data using MSE, MAE, and RMSE.(c)Algorithm-specific evaluation results according to Table 7 are as follows:i.ARIMA: with results of MSE (0.144), MAE (0.253), and RMSE (0.380), demonstrating performance comparable to SARIMA.ii.SARIMA: exhibiting MSE (0.143), MAE (0.250), and RMSE (0.378), indicating performance second only to SVM.iii.LSTM: showing the highest error values with MSE (0.675), MAE (0.777), and RMSE (0.821), indicating the lowest prediction accuracy for this dataset.iv.SVM: achieving the lowest MSE (0.076), MAE (0.198), and RMSE (0.276), signifying the most accurate predictions for this dataset.2.Consumer number (CNSMR_NO): DJ0800133001204.(a)Figure 6: graphs illustrating the differences between algorithm-specific predicted and actual data.
Figure 6Differences in SGPowerUsage values between predictions and actual data in DJ0800133001204.
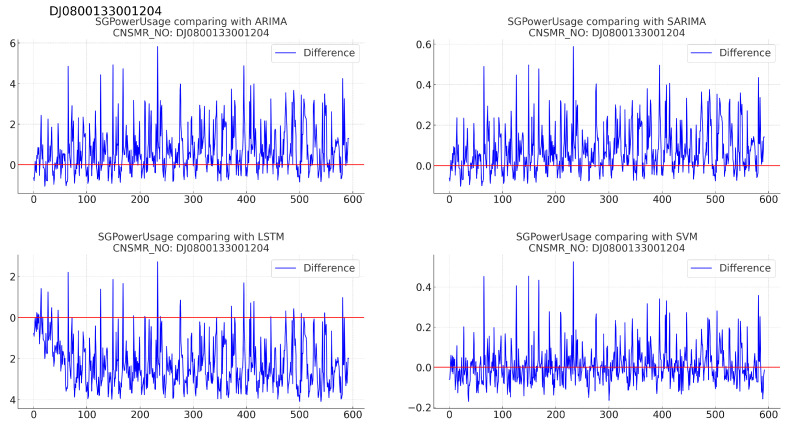
(b)Table 8: accuracy evaluation for the predicted data using MSE, MAE, and RMSE for each algorithm.(c)Evaluation results for each algorithm based on Table 8 are as follows:i.ARIMA: with MSE (0.018), MAE (0.094), and RMSE (0.134), showing similar performance to SARIMA.ii.SARIMA: shows slightly higher error rates than ARIMA with MSE (0.019), MAE (0.097), and RMSE (0.138), positioning it as second-best after SVM.iii.LSTM: recorded the highest error rates with MSE (0.065), MAE (0.232), and RMSE (0.255), indicating the lowest prediction accuracy for this dataset.iv.SVM: exhibited the lowest error rates with MSE (0.009), MAE (0.068), and RMSE (0.097), signifying the most accurate predictions for this dataset.3.Consumer number (CNSMR_NO): DJ1200215000404.(a)Figure 7: graphs illustrating the differences between algorithm-specific predicted and actual data.
Figure 7Differences in SGPowerUsage values between predictions and actual data in DJ1200215000404.
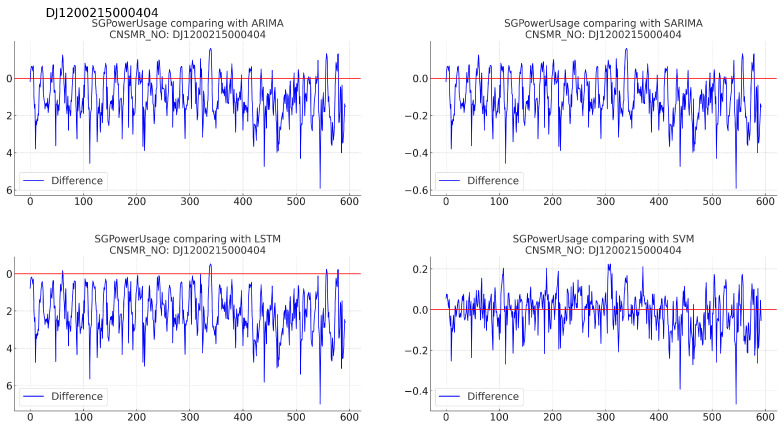
(b)Table 9: accuracy evaluation for the predicted data using MSE, MAE, and RMSE for each algorithm.(c)Evaluation results for each algorithm based on Table 9 are as follows:i.ARIMA: demonstrates similar performance to SARIMA with MSE (0.024), MAE (0.124), and RMSE (0.154), indicating closely matching results down to several decimal places.ii.SARIMA: exhibits comparable results to ARIMA, positioning it as effective after SVM in terms of performance.iii.LSTM: shows the highest error rates with MSE (0.057), MAE (0.208), and RMSE (0.238), indicating the lowest prediction accuracy for this dataset.iv.SVM: achieves the lowest error rates with MSE (0.009), MAE (0.071), and RMSE (0.093), signifying the most accurate predictions for this dataset.4.Consumer number (CNSMR_NO): CB0100106000505.(a)Figure 8: graphs illustrating the differences between algorithm-specific predicted and actual data.
Figure 8Differences in SGPowerUsage values between predictions and actual data in CB0100106000505.
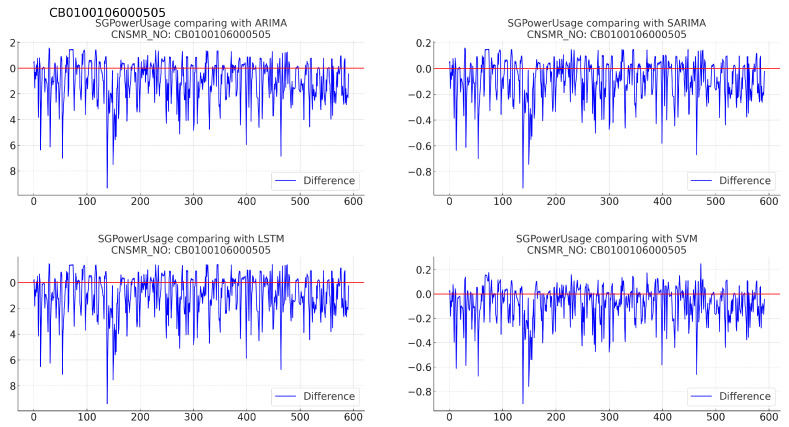
(b)Table 10: accuracy evaluation for the predicted data using MSE, MAE, and RMSE for each algorithm.(c)Evaluation results for each algorithm based on Table 10 are as follows:i.ARIMA: recorded the highest error with MSE (0.034), MAE (0.133), and RMSE (0.185).ii.SARIMA: demonstrates slightly better performance than ARIMA with MSE (0.032), MAE (0.130), and RMSE (0.179), making it second-best after SVM.iii.LSTM: shows marginally higher error than SARIMA with MSE (0.034), MAE (0.132), and RMSE (0.184) for this dataset.iv.SVM: exhibited the lowest error rates with MSE (0.029), MAE (0.117), and RMSE (0.169), indicating the most accurate predictions for this dataset.5Consumer number (CNSMR_NO): CN0100107001801.(a)Figure 9: graphs illustrating the differences between algorithm-specific predicted and actual data.
Figure 9Differences in SGPowerUsage values between predictions and actual data in CN0100107001801.
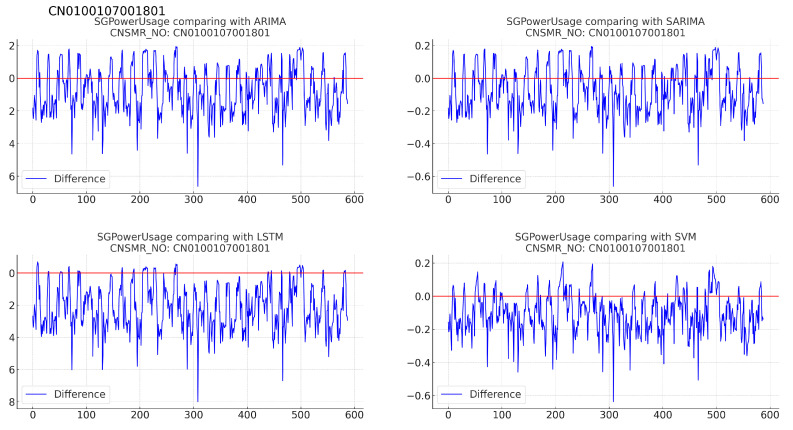
(b)Table 11: accuracy evaluation for the predicted data using MSE, MAE, and RMSE for each algorithm.(c)Evaluation results for each algorithm based on Table 11 are as follows:i.ARIMA: exhibited MSE (0.026), MAE (0.132), and RMSE (0.160), showing very similar performance to SARIMA.ii.SARIMA: recorded MSE (0.025), MAE (0.132), and RMSE (0.160), providing the most accurate predictions for this dataset.iii.LSTM: demonstrated the highest error rates with MSE (0.065), MAE (0.216), and RMSE (0.254) for this dataset.iv.SVM: showed slightly higher error rates with MSE (0.026), MAE (0.132), and RMSE (0.161) compared with ARIMA and SARIMA, but remained close in prediction accuracy.6Consumer number (CNSMR_NO): CN0200311001801.(a)Figure 10: graphs illustrating the differences between algorithm-specific predicted and actual data.
Figure 10Differences in SGPowerUsage values between predictions and actual data in CN0200311001801.
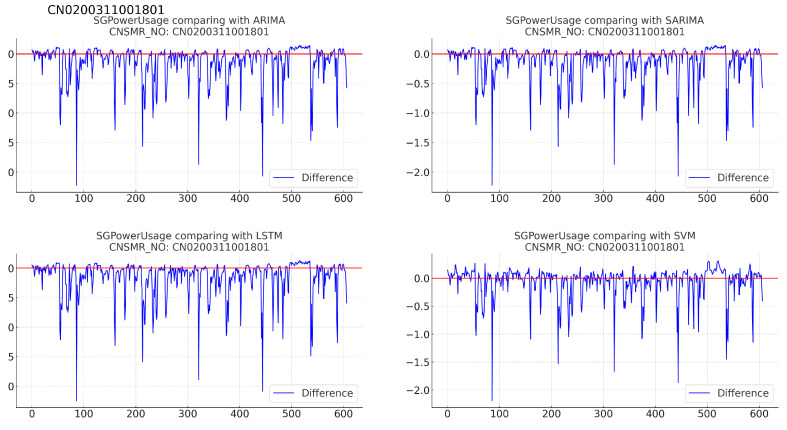
(b)Table 12: accuracy evaluation for the predicted data using MSE, MAE, and RMSE for each algorithm.(c)Evaluation results for each algorithm based on Table 12 are as follows:i.ARIMA: displayed MSE (0.111), MAE (0.178), and RMSE (0.333), showing nearly identical performance to SARIMA.ii.SARIMA: recorded MSE (0.111), MAE (0.178), and RMSE (0.333), demonstrating performance second only to SVM.iii.LSTM: had the highest error rates with MSE (0.117), MAE (0.185), and RMSE (0.342) for this dataset.iv.SVM: achieved the lowest error rates with MSE (0.087), MAE (0.161), and RMSE (0.295), indicating the most accurate predictions for this dataset.7.Consumer number (CNSMR_NO): CN1100106000103.(a)Figure 11: graphs illustrating the differences between algorithm-specific predicted and actual data.
Figure 11Differences in SGPowerUsage values between predictions and actual data in CN1100106000103.
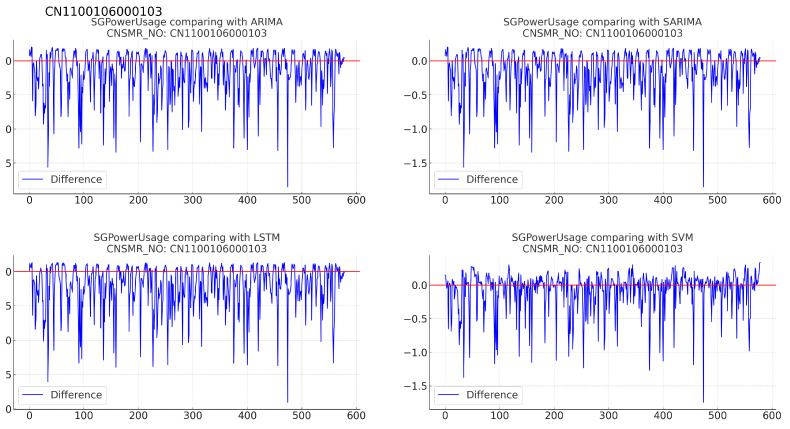
(b)Table 13: accuracy evaluation for the predicted data using MSE, MAE, and RMSE for each algorithm.(c)Evaluation results for each algorithm based on Table 13 are as follows:i.ARIMA: demonstrated MSE (0.136), MAE (0.242), RMSE (0.368).ii.SARIMA: showed identical performance to ARIMA with MSE (0.136), MAE (0.242), RMSE (0.368).iii.LSTM: recorded the highest error rates with MSE (0.156), MAE (0.255), RMSE (0.395) in this dataset.iv.SVM: achieved the lowest error rates with MSE (0.096), MAE (0.187), RMSE (0.310), providing the most accurate predictions for this dataset.8.Consumer number (CNSMR_NO): CN1600102001004.(a)Figure 12: graphs illustrating the differences between algorithm-specific predicted and actual data.
Figure 12Differences in SGPowerUsage values between predictions and actual data in CN1600102001004.
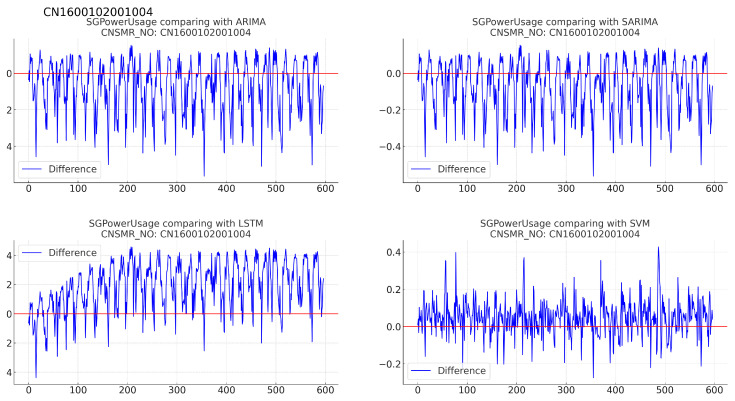
(b)Table 14: accuracy evaluation for the predicted data using MSE, MAE, and RMSE for each algorithm.(c)Evaluation results for each algorithm based on Table 14 are as follows:i.ARIMA: reported MSE (0.026), MAE (0.122), and RMSE (0.162).ii.SARIMA: exhibited identical performance to ARIMA with MSE (0.026), MAE (0.122), and RMSE (0.162).iii.LSTM: recorded the highest error rates with MSE (0.063), MAE (0.216), and RMSE (0.251) for this dataset.iv.SVM: achieved the lowest error rates with MSE (0.011), MAE (0.082), and RMSE (0.106), providing the most accurate predictions for this dataset.9.Consumer number (CNSMR_NO): CN0500311000403.(a)Figure 13: graphs illustrating the differences between algorithm-specific predicted and actual data.
Figure 13Differences in SGPowerUsage values between predictions and actual data in CN0500311000403.
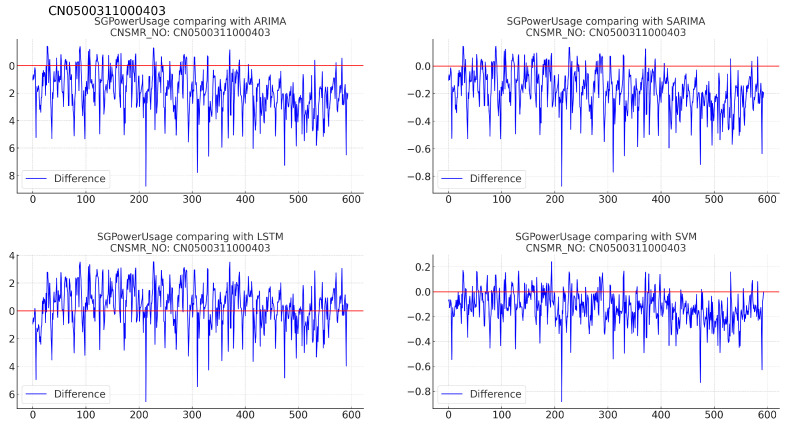
(b)Table 15: accuracy evaluation for the predicted data using MSE, MAE, and RMSE for each algorithm.(c)Evaluation results for each algorithm based on Table 15 are as follows:i.ARIMA: exhibited MSE (0.057), MAE (0.190), and RMSE (0.232), indicating the highest error values for this dataset.ii.SARIMA: reported MSE (0.054), MAE (0.190), and RMSE (0.232), showing slightly higher errors compared with SVM.iii.LSTM: achieved the lowest error rates with MSE (0.024), MAE (0.122), and RMSE (0.156), providing the most accurate predictions for this dataset.iv.SVM: recorded MSE (0.034), MAE (0.147), and RMSE (0.184), demonstrating the second-best performance after LSTM.10.Consumer number (CNSMR_NO): CN0700109000102.(a)Figure 14: graphs illustrating the differences between algorithm-specific predicted and actual data.
Figure 14Differences in SGPowerUsage values between predictions and actual data in CN0700109000102.
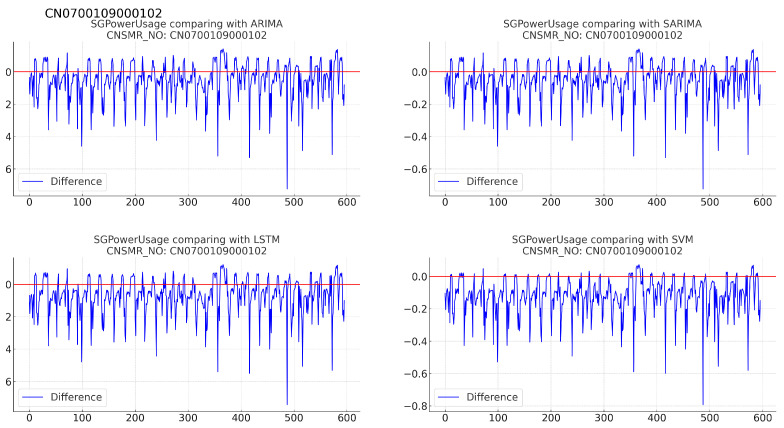
(b)Table 16: accuracy evaluation for the predicted data using MSE, MAE, and RMSE for each algorithm.(c)Evaluation results for each algorithm based on Table 16 are as follows:i.ARIMA: achieved MSE (0.014), MAE (0.085), RMSE (0.120), providing the most accurate predictions for this dataset.ii.SARIMA: recorded identical performance to ARIMA with MSE (0.014), MAE (0.085), RMSE (0.120).iii.LSTM: showed slightly higher error rates with MSE (0.017), MAE (0.095), RMSE (0.130) compared with ARIMA and SARIMA.iv.SVM: reported the highest error rates with MSE (0.026), MAE (0.124), RMSE (0.161) for this dataset.

Figure 15 illustrates the overall differences in SGPowerUsage values between the predictions and actual data for each model (SVM,LSTM,SARIMA,ARIMA). For reference, each graph includes a red dotted zero line. Points above this line indicate an overestimation by the model, while points below indicate an underestimation.

1.ARIMA: exhibits performance similar to SARIMA but with a slightly higher MAE.2SARIMA: generally closer to the zero line compared with LSTM, though with some deviations, which are reflected in the MAE.3.LSTM: shows larger deviations from the zero line, resulting in a higher MAE compared with other models.4.SVM: demonstrates relatively small fluctuations around the zero line, indicating predictions closer to the actual values, consistent with the lowest MAE.

Based on the overall experimental results, it can be concluded that SVM is the most accurate algorithm for scenarios like short-term forecasting in AMI, where the dataset is small or limited resources are available for prediction.

## 5. Discussion

This study evaluated the performance of four predictive models, namely SVM, ARIMA, SARIMA, and LSTM, using real-world AMI (Advanced Metering Infrastructure) data. The main objective was to identify which algorithm is most effective for short-term electricity demand forecasting based on AMI data. The evaluation was conducted using statistical metrics such as Mean Squared Error (MSE), Mean Absolute Error (MAE), and Root Mean Squared Error (RMSE).

### 5.1. Expected Model Performance

Initially, the expected ranking of the algorithms was as follows:1.LSTM: Due to its ability to learn long-term dependencies and model nonlinear patterns, LSTM was expected to perform best, especially for large datasets. Its capacity to handle sequential data and complex patterns made it a prime candidate for time series forecasting [32,33].2SVM: SVM was anticipated to be effective for capturing specific nonlinear patterns through the use of various kernel functions. Although slower for larger datasets, its flexibility in transforming data into higher-dimensional spaces made it a strong competitor in predicting nonlinear trends [34,35].3.SARIMA: Specialized in handling seasonality in time series data, SARIMA was expected to model seasonal patterns accurately. However, it may have been limited when dealing with complex nonlinear interactions [36].4.ARIMA: ARIMA, as a linear model, was expected to perform well for datasets with clear linear trends but might struggle with seasonality or nonlinearities [37].

### 5.2. Observed Model Performance and Insights

Contrary to the initial expectations, the results showed that the Support Vector Machine (SVM) model demonstrated the most consistent and superior performance across the majority of the datasets by achieving the lowest error metrics. This suggests that SVM’s strength lies in its ability to generalize effectively and capture complex, nonlinear patterns through the use of kernel functions. SVM’s flexibility in high-dimensional spaces enabled it to outperform other models, particularly in scenarios with intricate data structures [38].

In contrast, the ARIMA and SARIMA models were effective at modeling linear trends and seasonal variations in the data. However, their performance was comparatively weaker due to their limited ability to model nonlinear patterns. This is consistent with the understanding that while traditional linear models are strong at capturing deterministic patterns, they often underperform when faced with complex temporal dependencies and nonlinearity [39].

The LSTM model exhibited highly variable performance across datasets. Although LSTM is theoretically powerful for capturing long-term dependencies and sequential patterns, its effectiveness in practice depends heavily on the quality and quantity of training data, as well as careful tuning of hyperparameters [40,41,42]. This indicates that LSTM might require a more extensive training period and larger datasets to fully realize its potential, which may not have been feasible given the data limitations in this study.

### 5.3. Considerations for AMI Data and Resource Constraints

When applying these models in AMI-based short-term forecasting, it is crucial to consider not only prediction accuracy but also computational efficiency, memory management, and training time. Each model has unique strengths and limitations in terms of resource requirements:ARIMA and SARIMA: These models are relatively simple to implement and interpret, making them suitable for scenarios with limited computational resources. However, their parameter estimation can become time-consuming as the data size increases.LSTM: while capable of capturing complex patterns, LSTM requires a substantial amount of training data and computational power, making it less practical for smaller datasets or resource-constrained environments.SVM: SVM performs well with small to medium-sized datasets and is particularly effective at handling nonlinearities. However, its computational complexity increases significantly with larger datasets, which can pose challenges in high-dimensional spaces.

### 5.4. Experimental Scope and Limitations

In this study, we conducted experiments using data from 10 residential households. While we recognize that a larger sample size could yield more generalizable results, the decision to focus on 10 households was based on several practical considerations. Firstly, data collection and processing from Advanced Metering Infrastructure (AMI) systems is resource-intensive, requiring significant effort to ensure accuracy and reliability. Secondly, the primary goal of this study is not to generalize the findings to all residential consumers but to compare the performance of various time series forecasting models in a controlled setting. The insights derived from these 10 households provide a valuable foundation for future research that can be expanded to larger datasets.

Although the limited sample size may affect the generalizability of the results, the study’s primary objective is to evaluate the comparative performance of forecasting algorithms within a specific context. The findings still offer meaningful insights into the applicability of these models for short-term load forecasting in AMI-based demand response systems.

### 5.5. Data Period and Time Horizon Considerations

The data used in this study span a 5-month period from July 2021 to December 2021. This timeframe was selected to capture both high-demand periods, such as the summer months, and lower-demand periods during the fall and early winter. While a full 12-month dataset would provide a more comprehensive view of seasonal variations, the 5-month period offers sufficient data to evaluate the short-term forecasting capabilities of the models under study. Additionally, this timeframe encompasses a range of consumption behaviors, providing diverse data for model evaluation.

We acknowledge that the use of a 5-month dataset may limit the ability to fully capture long-term seasonal trends. However, given the study’s focus on short-term load forecasting, the chosen time horizon is appropriate for assessing the performance of forecasting algorithms within a shorter, operationally relevant timeframe.

## 6. Conclusions

This study systematically compared four time series forecasting models—ARIMA, SARIMA, LSTM, and SVM—using the same AMI dataset to evaluate their effectiveness for short-term electricity demand forecasting in demand response (DR) applications. The objective was to identify the most suitable model for capturing the unique characteristics of AMI data, such as high variability, seasonality, and nonlinear patterns, and to provide practical insights for implementing these models in real-world AMI systems.

The findings indicate that SVM consistently outperformed the other models, achieving the lowest error metrics and demonstrating a robust ability to capture complex nonlinear patterns and adapt to dynamic changes in energy usage. This makes SVM the most appropriate model for short-term forecasting in AMI-based DR programs, where accurate and timely predictions are essential. However, it is important to note that these results are based on a specific dataset with a limited number of households, which may not fully capture the diversity of electricity consumption patterns observed in larger populations.

ARIMA and SARIMA were effective at modeling linear trends and seasonal patterns, making them reliable choices for scenarios with predictable and stable energy consumption trends. However, their limited capacity to handle nonlinearities and abrupt changes resulted in higher error metrics in more volatile datasets, which may limit their application in dynamic AMI-based DR programs.

LSTM, while theoretically advantageous due to its capacity to model long-term dependencies and complex temporal patterns, showed inconsistent performance across the dataset. Its effectiveness depends heavily on the availability of large-scale training data and the optimization of hyperparameters, which may not always be feasible in AMI environments with limited data and resources.

### 6.1. Contributions and Practical Implications

This study offers several key contributions:Comprehensive model evaluation: this research provides an in-depth performance evaluation of four widely used forecasting models, offering a clear understanding of each model’s strengths and limitations under identical conditions.Guidelines for model selection in AMI systems: The findings suggest that SVM is the most suitable choice for capturing complex patterns in AMI data, while ARIMA and SARIMA are better suited for linear and seasonal trends. LSTM should only be considered when extensive data and computational resources are available.Foundation for future research: the study establishes a baseline for further exploration of hybrid models and advanced forecasting techniques, contributing to more effective implementation of DR strategies in smart grid environments.

### 6.2. Recommendations for Future Research

Future research should focus on the following:Real-world AMI data validation: testing these models on real-world AMI data from various regions and consumer segments to validate their effectiveness under diverse operational conditions and to ensure the results are generalizable to a broader set of AMI systems.Exploring hybrid models: developing hybrid models that combine the strengths of multiple algorithms, such as SARIMA + SVM or LSTM + SVM, to improve prediction accuracy and robustness in more complex AMI environments.Resource optimization for scalability: optimizing the computational efficiency and scalability of these models to ensure their feasibility in large-scale AMI deployments with resource constraints.

Overall, SVM is recommended as the most effective model for short-term forecasting in AMI systems with similar characteristics and scale to the dataset used in this study. However, the application of these findings to larger and more diverse datasets should be approached with caution. ARIMA and SARIMA remain reliable for stable, linear trends, while LSTM’s application should be limited to cases where substantial data and computational resources are available. These findings provide practical guidelines for utility companies and researchers seeking to implement efficient demand response strategies based on accurate short-term forecasting.

## Figures and Tables

**Figure 1 sensors-24-07205-f001:**
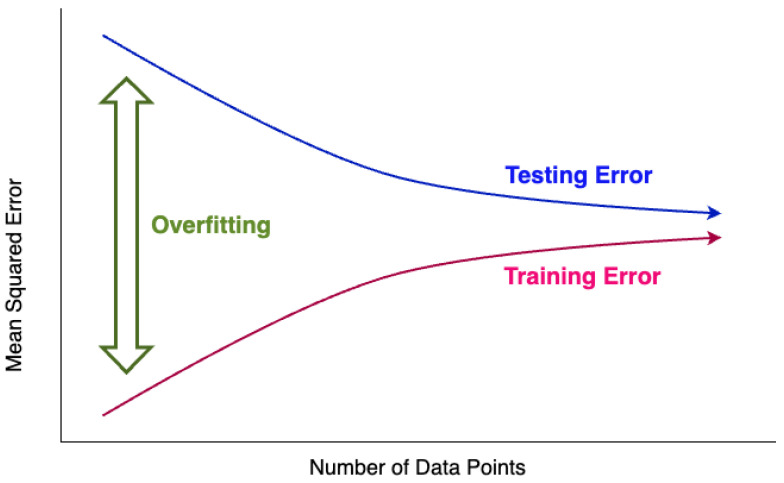
Possibility of overfitting due to amount of learning data.

**Figure 2 sensors-24-07205-f002:**
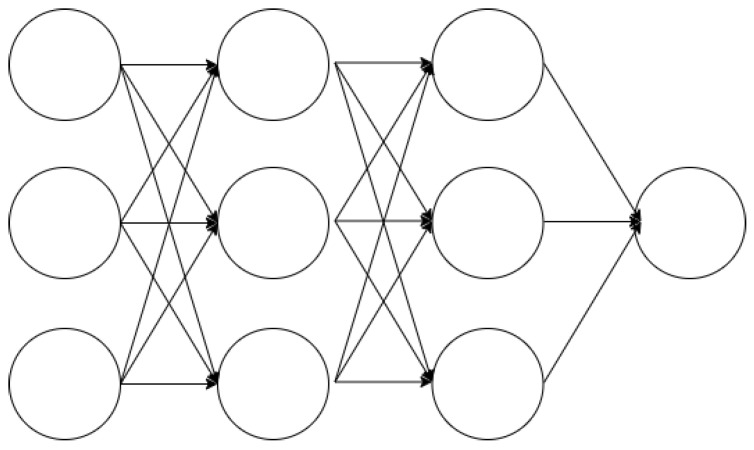
Neural network without dropout.

**Figure 3 sensors-24-07205-f003:**
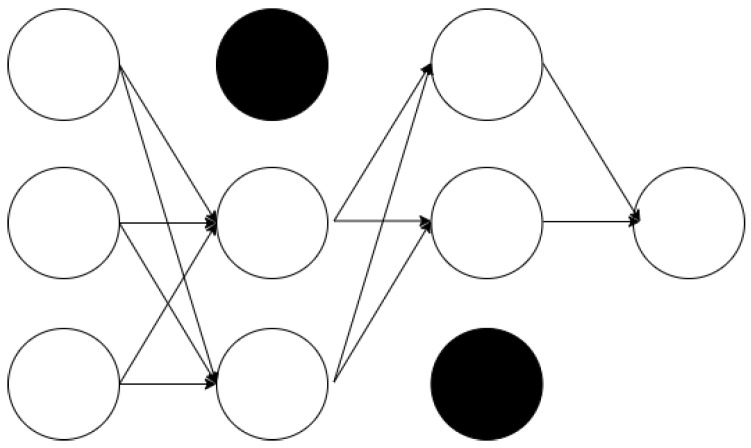
Neural network with dropout.

**Figure 4 sensors-24-07205-f004:**
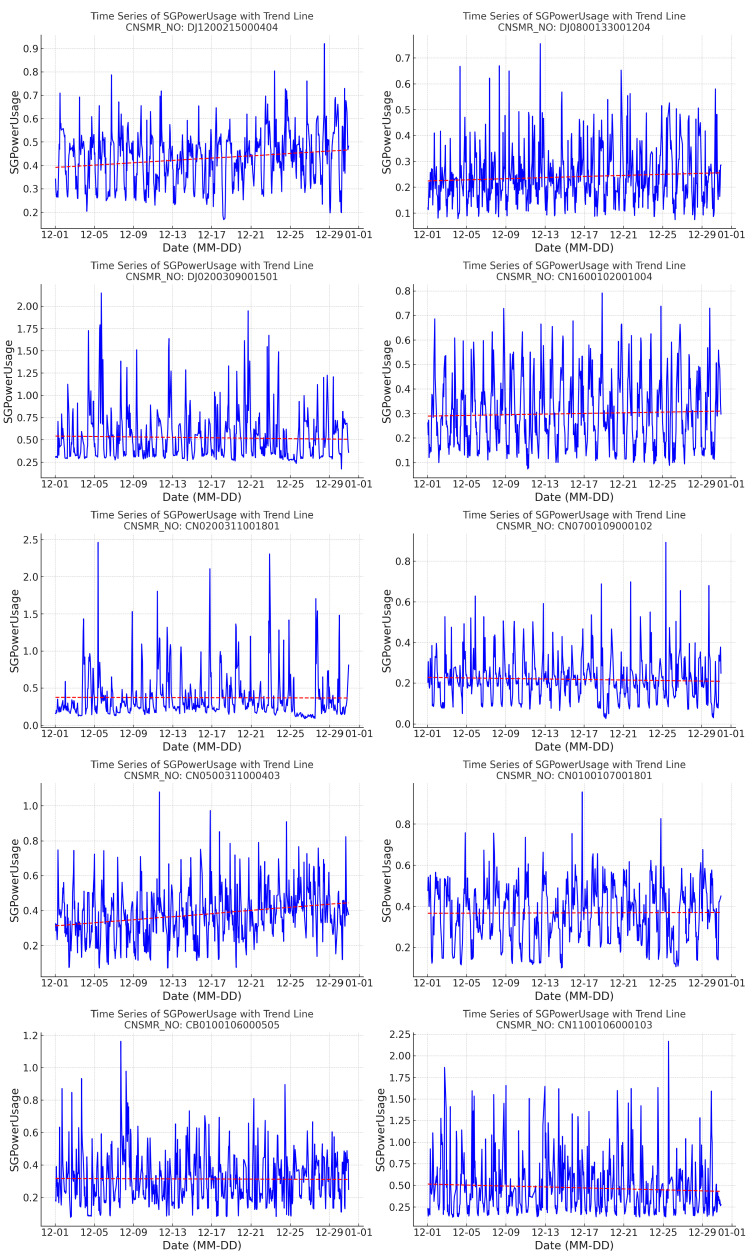
Distribution and trend line graphs of sampled data by consumer number (CNSMR_NO).

**Figure 15 sensors-24-07205-f015:**
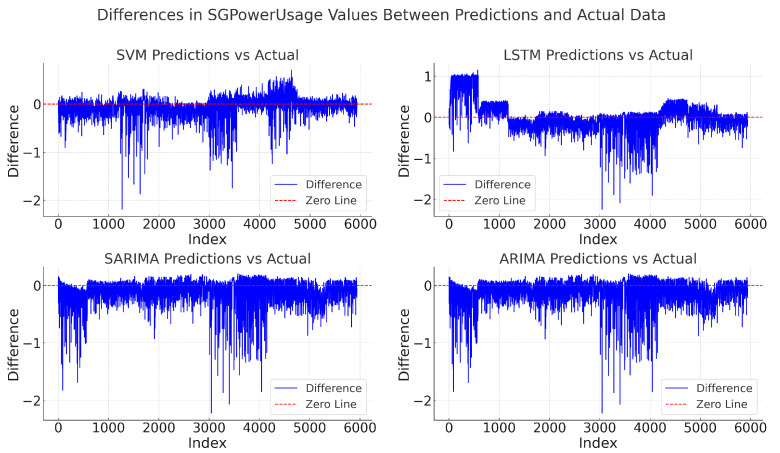
Differences in SGPowerUsage values between predictions and actual data.

**Table 1 sensors-24-07205-t001:** Data specification for time series analysis.

English Name	Type	Length	Required	Identifier	Allowed Values	Example
TOC_NO	VARCHAR	10	Y	Y	Alphabet followed by numerical string	A000000001
CNSMR_NO	VARCHAR	20	Y	Y	Random string	KB0100201000105
LEGALDONG_CD	VARCHAR	10	Y	-	Numerical string	2653010400
MESURE_DE	VARCHAR	8	Y	Y	Post-2016	20190901
MESURE_TM	VARCHAR	2	Y	Y	00 23	14

**Table 2 sensors-24-07205-t002:** Data specification for time series learning and analysis.

English Name	Type	Length	Required	Identifier	Example	
CNSMR_NO	VARCHAR	20	Y	Y	Random string	KB0100201000105
LEGALDONG_CD	VARCHAR	10	Y		Number	2653010400
MESURE_YEAR	VARCHAR	4	Y	Y	After 2016	2019
MESURE_MT	VARCHAR	2	Y	Y	Number	01
MESURE_DTTM	VARCHAR	2	Y	Y	Number	31
MESURE_TM	VARCHAR	2	Y	Y	00 23	14
HLDY_AT	VARCHAR	1	Y		Y/N	Y
SG_PWRER_ USE_AM	NUMBER	16	Y		0.0 9999999999.9999	0.345

**Table 3 sensors-24-07205-t003:** Slope and intercept of forecasting data using ARIMA (each CNSMR_NO).

CNSMR_NO	Slope	Intercept
CB0100106000505	−0.001011758	20,449.089
CN0100107001801	−9.081498547	0.313582719
CN0200311001801	2.113108398	−426.846467
CN0500311000403	−0.00149791	30,274.77219
CN0700109000102	−5.007757692	1.181070942
CN1100106000103	−4.210372471	851.2864752
CN1600102001004	8.987788029	−17.93733817
DJ0200309001501	−0.005480365	110,765.1287
DJ0800133001204	−0.000982641	19,860.54595
DJ1200215000404	5.463169948	−110.0872162

**Table 4 sensors-24-07205-t004:** Slope and intercept of forecasting data using SARIMA (each CNSMR_NO).

CNSMR_NO	Slope	Intercept
CB0100106000505	−0.000319073	6449.082179
CN0100107001801	4.039360831	−0.520846808
CN0200311001801	2.280846442	−460.7483028
CN0500311000403	−0.001123215	22,701.73422
CN0700109000102	−5.009475262	1.1814181
CN1100106000103	−4.210357905	851.2835312
CN1600102001004	8.987780947	−17.93732388
DJ0200309001501	−0.005493802	11,1036.6921
DJ0800133001204	−0.001262886	25,524.63396
DJ1200215000404	5.463022253	−110.0842315

**Table 5 sensors-24-07205-t005:** Slope and intercept of forecasting data using LSTM (each CNSMR_NO).

CNSMR_NO	Slope	Intercept
CB0100106000505	5.580297358	−1127.623133
CN0100107001801	−0.000230448	4657.795984
CN0200311001801	−4.785285336	967.3830755
CN0500311000403	0.001596365	−32,264.05829
CN0700109000102	9.239923789	−1867.352245
CN1100106000103	−2.525308407	510.6578743
CN1600102001004	0.007655379	−154724.0285
DJ0200309001501	0.007879234	−159,247.6266
DJ0800133001204	0.003431755	−69,359.46656
DJ1200215000404	−0.000103266	2087.36154

**Table 6 sensors-24-07205-t006:** Slope and intercept of forecasting data using SVM (each CNSMR_NO).

CNSMR_NO	Slope	Intercept
CB0100106000505	8.233022059	0.2380441
CN0100107001801	8.233022059	0.262161614
CN0200311001801	−7.409719853	0.309545457
CN0500311000403	1.646604412	0.258864742
CN0700109000102	0	0.1
CN1100106000103	−3.293208824	0.407003965
CN1600102001004	2.140585735	0.357806174
DJ0200309001501	3.293208824	0.651065178
DJ0800133001204	−6.586417647	0.238392017
DJ1200215000404	1.317283529	0.424890753

**Table 7 sensors-24-07205-t007:** DJ0200309001501 performance evaluation.

Consumer Number (CNSMR_NO)	DJ0200309001501		
Algorithm	MSE	MAE	RMSE
ARIMA	0.144	0.253	0.38
SARIMA	0.143	0.25	0.378
LSTM	0.675	0.777	0.821
SVM	0.076	0.198	0.276

**Table 8 sensors-24-07205-t008:** DJ0800133001204 performance evaluation.

Consumer Number (CNSMR_NO)	DJ0800133001204		
Algorithm	MSE	MAE	RMSE
ARIMA	0.018	0.094	0.134
SARIMA	0.019	0.097	0.138
LSTM	0.065	0.232	0.255
SVM	0.009	0.068	0.097

**Table 9 sensors-24-07205-t009:** DJ1200215000404 performance evaluation.

Consumer Number (CNSMR_NO)	DJ1200215000404		
Algorithm	MSE	MAE	RMSE
ARIMA	0.024	0.124	0.154
SARIMA	0.024	0.124	0.154
LSTM	0.057	0.208	0.238
SVM	0.009	0.071	0.093

**Table 10 sensors-24-07205-t010:** CB0100106000505 performance evaluation.

Consumer Number (CNSMR_NO)	CB0100106000505		
Algorithm	MSE	MAE	RMSE
ARIMA	0.034	0.133	0.185
SARIMA	0.032	0.130	0.179
LSTM	0.034	0.132	0.184
SVM	0.029	0.117	0.169

**Table 11 sensors-24-07205-t011:** CN0100107001801 performance evaluation.

Consumer Number (CNSMR_NO)	CN0100107001801		
Algorithm	MSE	MAE	RMSE
ARIMA	0.026	0.132	0.160
SARIMA	0.025	0.132	0.160
LSTM	0.065	0.216	0.254
SVM	0.026	0.132	0.161

**Table 12 sensors-24-07205-t012:** CN0200311001801 performance evaluation.

Consumer Number (CNSMR_NO)	CN0200311001801		
Algorithm	MSE	MAE	RMSE
ARIMA	0.111	0.178	0.333
SARIMA	0.111	0.178	0.333
LSTM	0.117	0.185	0.342
SVM	0.087	0.161	0.295

**Table 13 sensors-24-07205-t013:** CN1100106000103 performance evaluation.

Consumer Number (CNSMR_NO)	CN1100106000103		
Algorithm	MSE	MAE	RMSE
ARIMA	0.136	0.242	0.368
SARIMA	0.136	0.242	0.368
LSTM	0.156	0.255	0.395
SVM	0.096	0.187	0.310

**Table 14 sensors-24-07205-t014:** CN1600102001004 performance evaluation.

Consumer Number (CNSMR_NO)	CN1600102001004		
Algorithm	MSE	MAE	RMSE
ARIMA	0.026	0.122	0.162
SARIMA	0.026	0.122	0.162
LSTM	0.063	0.216	0.251
SVM	0.011	0.082	0.106

**Table 15 sensors-24-07205-t015:** CN0500311000403 performance evaluation.

Consumer Number (CNSMR_NO)	CN0500311000403		
Algorithm	MSE	MAE	RMSE
ARIMA	0.057	0.190	0.232
SARIMA	0.054	0.190	0.232
LSTM	0.024	0.122	0.156
SVM	0.034	0.147	0.184

**Table 16 sensors-24-07205-t016:** CN0700109000102 performance evaluation.

Consumer Number (CNSMR_NO)	CN0700109000102		
Algorithm	MSE	MAE	RMSE
ARIMA	0.014	0.085	0.120
SARIMA	0.014	0.085	0.120
LSTM	0.017	0.095	0.130
SVM	0.026	0.124	0.161

## Data Availability

The source code for this study is available at https://github.com/taesachi/short-term-forecast-dr (accessed on 28 January 2024). As for the dataset, it is provided by KT and can be accessed through their platform at https://www.bigdata-telecom.kr/ (accessed on 28 January 2024). To obtain the data, users must complete the data request process on the website, after which the dataset will be delivered via email. Due to these procedures, we are unable to provide the dataset directly. If you wish to use the data, we kindly ask that you request it directly from KT through their official platform.

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
