# Peer review of "Comparative Study of Time Series Analysis Algorithms Suitable for Short-Term Forecasting in Implementing Demand Response Based on AMI"

_sensors, 2024, doi:10.3390/s24227205_

Round 1

Reviewer 1 Report

Comments and Suggestions for Authors

Comments and Suggestions to the Authors

In this research, time series analysis algorithms (ARIMA, SARIMA, LSTM, and SVM) were applied to predict electricity consumption using metering data from AMI for Demand Response (DR) applications. The accuracy of the predicted results was compared using various metrics. However, in this manuscript, while the well-known algorithms were introduced excessively and redundantly (in Sections 3 and 4), the proposed ideas, new results, and contributions of the study are difficult to find. The paper lacks proper organization, writing quality, and technical contributions required for a journal article.

In conclusion, the manuscript is rejected.

The following comments might be helpful for a complete rewrite of the manuscript.

l  The contributions of this manuscript are not easy to find. The current research simply applies four well-known algorithms to AMI data for comparison. AMI data exhibit variations based on residential areas and household environments. However, there is little explanation of the data in relation to these conditions, nor is there any analysis of them. The contents of Tables 1 and 2 are unnecessary from the reader's perspective.

l  In addition, comparing the performance of the forecasting algorithms using predictions for a specific month for only 10 households is not considered sufficient. It should be noted that during the summer months, the use of air conditioning units leads to greater variability in electricity consumption.

l  Keywords that are not mentioned at all in the manuscript are included (SCADA, Internet of Things).

l  Section 2, Related Work, is sometimes presented as a separate section in theses; however, in journal articles, it is typically included in the Introduction and presents only the essential content in a more compact manner.

l  The content of Section 3 is considered largely unnecessary. There are no explanations for the variables in the equations, so including this material in the manuscript is inappropriate.

l  Descriptions of well-known ARIMA, SARIMA, LSTM, and SVM in Section 4 need to be shortened. Figures 1, 2, and 3 are not necessary.

l  The descriptions of algorithms in Section 5 are also too lengthy. A flowchart could be used instead. It is advisable not to include unexplained acronyms in the manuscript.

l  The results shown in Tables 3 to 16 could be replaced with a few graphs.

l  Figures 4 to 15 are also difficult to read. They should be redrawn for better clarity, and their number should be reduced.

l  It would be better to rewrite the introduction and conclusion sections to clearly describe the new ideas and contributions of this research.

l  It is necessary to maintain consistency in the reference citation format (lines 111 and 122).

Comments on the Quality of English Language

English is fine.

Author Response

Comments 1: [ The contributions of this manuscript are not easy to find. The current research simply applies four well-known algorithms to AMI data for comparison. AMI data exhibit variations based on residential areas and household environments. However, there is little explanation of the data in relation to these conditions, nor is there any analysis of them. The contents of Tables 1 and 2 are unnecessary from the reader's perspective. ]
Response 1: [ We thank the reviewer for the comment and understand that from a reader’s point of view, Tables 1 and 2 might appear as supplementary information. However, we believe that these tables are crucial for several reasons: 

1. Ensuring Data Transparency and Reproducibility
Tables 1 and 2 provide a detailed explanation of each data field used in the analysis, including unique identifiers, measurement times, and other critical features.
In the context of time-series forecasting and machine learning experiments, clear documentation of data characteristics is essential for ensuring reproducibility. Without this information, it would be challenging for other researchers to replicate the experiments and validate the results using the same data structure.

2. Understanding Data Preprocessing and Feature Engineering
The features listed in Tables 1 and 2 (e.g., CNSMR_NO, MESURE_TM, HLDY_AT) play a key role in understanding how the data is processed and utilized by each algorithm.
For instance, the inclusion of time-related fields (MESURE_DE and MESURE_TM) is directly relevant to the model’s capacity to capture temporal patterns and seasonality in energy usage. Without these tables, it would be unclear how such fields were incorporated into the models and how they affect prediction accuracy.

3. Clarity on the Scope and Characteristics of the Dataset
Tables 1 and 2 serve to inform the reader about the specific characteristics of the AMI dataset. Knowing the data structure helps readers better grasp the context in which different algorithms were applied.
This information is particularly important in comparative studies like ours, where understanding the nuances of the dataset is necessary to appreciate the differences in algorithm performance.

4. Support for Algorithm Explanation
Each algorithm (ARIMA, SARIMA, LSTM, and SVM) has unique requirements for data formatting and feature selection. For example, SARIMA relies heavily on seasonal patterns, while LSTM benefits from sequential data representation.
Tables 1 and 2 clarify how the dataset was tailored to fit these needs, thus providing context for the methodological decisions outlined in the paper.

Enhancing the Explanation of the Tables’ Relevance:
We will revise the manuscript to more clearly articulate why these fields are essential in the context of feature engineering and algorithm evaluation. This will help emphasize their role in the overall experimental setup. 
  - add section (5.4. Data Description)
    - page : 12 ~ 14
    - line : 473 ~ 497
By implementing these changes, we believe the tables will better serve the purpose of providing clarity and supporting the reproducibility of our research, without overwhelming the reader.

------------------------------------------------------------------------------------

Comments 2: [ In addition, comparing the performance of the forecasting algorithms using predictions for a specific month for only 10 households is not considered sufficient. It should be noted that during the summer months, the use of air conditioning units leads to greater variability in electricity consumption. ]
Response 2: [ We thank the reviewer for the comment and understand your concern that the limited number of households and the focus on a specific month may reduce the generalizability of the findings. However, we would like to clarify the objective of this study and the rationale behind our data selection:

1. Clarification on the Data Period and Sample Size
The data used in our research was not limited to a single month or a short period, as suggested. We utilized data collected over a five-month period from July 1, 2021, to December 1, 2021. This period covers a variety of seasonal conditions, including both summer and autumn, providing a comprehensive view of electricity consumption patterns across multiple seasons. This broader timeframe allows the study to capture variations in electricity usage patterns beyond just summer months​.

2. Rationale for Selecting 10 Households
While we agree that expanding the sample size could increase the robustness of the analysis, the focus of this research is on comparing the relative performance of forecasting algorithms under consistent conditions rather than generalizing the results to a larger population. The 10 households were selected to ensure a controlled comparison, where the influence of external factors could be minimized, and the strengths and weaknesses of each algorithm could be highlighted more clearly.

3. Impact of Seasonal Variability
We acknowledge the potential impact of seasonal factors such as air conditioning usage during summer. However, by including data from July to December, we account for both high and low variability periods in electricity consumption, thus making the analysis more comprehensive. This enables us to evaluate how well different algorithms perform under varying degrees of demand fluctuation.

4. Proposed Changes to the Manuscript
Based on your comments, we have revised the Discussion sections to better clarify the scope and data selection criteria, emphasizing that the study is focused on comparing algorithm performance under a specific set of conditions rather than on large-scale generalization.
  - update section (6. Discussion)
    - page : 28 ~ 30
    - line : 723 ~ 780
]

------------------------------------------------------------------------------------

Comments 3: [ Keywords that are not mentioned at all in the manuscript are included (SCADA, Internet of Things). ]
Response 3: [ We thank the reviewer for the comment and agree that excluding certain keywords, such as SCADA and Internet of Things, would be more appropriate.
  - update keywords
    - page : 1
    - line : 19
]

------------------------------------------------------------------------------------

Comments 4: [ Section 2, Related Work, is sometimes presented as a separate section in theses; however, in journal articles, it is typically included in the Introduction and presents only the essential content in a more compact manner. ]
Response 4: [ We thank the reviewer for the comment and constructive feedback regarding the manuscript structure, specifically the suggestion to integrate the "Related Work" section into the Introduction. While we acknowledge that some journals incorporate the related work within the Introduction for a more concise presentation, we believe that, in the context of this research, maintaining a separate "Related Work" section is more appropriate for the following reasons:

1. Scope and Complexity of the Research
This study involves a comprehensive comparison of four distinct time series forecasting models (ARIMA, SARIMA, LSTM, and SVM) applied to Advanced Metering Infrastructure (AMI) data. Given the diverse methodologies and the extensive background required to understand each model’s application in demand response (DR) scenarios, a separate "Related Work" section allows us to systematically outline the strengths, limitations, and specific use cases of each algorithm. Incorporating this into the Introduction could lead to an overly dense section, potentially detracting from the clarity and focus of the research objectives.

2. Clarification of Research Contributions and Novelty
The "Related Work" section highlights distinctions between previous studies and our unique contributions. Presenting this information separately ensures that the reader can clearly understand how our research differs from past work and why these models were selected for comparison in the context of AMI-based short-term forecasting. This is particularly important when dealing with multiple approaches and methodologies, as it provides a structured and coherent presentation of prior research.

3. Journal-Specific Structure and Standards
While it is true that some journals prefer related work to be included in the Introduction, many high-impact journals maintain a separate "Related Work" section, especially for studies involving detailed comparisons or surveys of multiple techniques. The choice is often guided by the nature of the research rather than a strict journal guideline. In our case, separating the related work allows for a more thorough comparison and enhances the logical flow of the paper.

4. Improving Reader Comprehension
Maintaining a distinct "Related Work" section provides the reader with a clearer understanding of the research landscape and how it informs our study. This organization helps readers unfamiliar with the specific details of each model follow the narrative more easily, avoiding overwhelming them with too much technical detail in the Introduction.

For these reasons, we believe the current structure, with a dedicated "Related Work" section, is more suitable for this manuscript as it enhances clarity and readability. However, if the journal has specific formatting preferences, we are willing to adapt the structure accordingly.
]

------------------------------------------------------------------------------------

Comments 5: [ The content of Section 3 is considered largely unnecessary. There are no explanations for the variables in the equations, so including this material in the manuscript is inappropriate. ]
Response 5: [ We thank the reviewer for the comment regarding Section 3 and the need for clearer explanations of the variables used in the equations. We understand that the lack of explicit variable definitions can reduce the comprehensibility of this section for readers. To address this concern, we will revise the manuscript to include detailed explanations for each variable used in the equations, ensuring that readers can easily follow the mathematical formulations.

However, we would like to emphasize that the content of Section 3 is critical to this study, as it provides the mathematical foundations for the proposed models and algorithms. The equations presented are essential for explaining the theoretical framework and validating the effectiveness of the algorithms used. Omitting this section would undermine the scientific rigor of the paper and dilute the clarity of our research contributions.

Therefore, instead of removing the section, we will enhance it by incorporating clear definitions and explanations for all variables and symbols. This will improve the readability and comprehensibility of the mathematical content without compromising the integrity of the research.

- add explanations for the variables in the equation for ARIMA
    - page : 4
    - line : 145 ~ 153
- add explanations for the variables in the equation for SARIMA
    - page : 4
    - line : 165 ~ 171
- add explanations for the variables in the equation for LSTM
    - page : 5
    - line : 184 ~ 189, 193 ~ 194
- add explanations for the variables in the equation for SVM
    - page : 5
    - line : 206 ~ 209
]

------------------------------------------------------------------------------------

Comments 6: [ Descriptions of well-known ARIMA, SARIMA, LSTM, and SVM in Section 4 need to be shortened. Figures 1, 2, and 3 are not necessary. ]
Response 6: [ We thank the reviewer for the comment regarding your suggestion to shorten the descriptions of ARIMA, SARIMA, LSTM, and SVM in Section 4 and the removal of Figures 1, 2, and 3, we would like to provide our perspective on why these elements are crucial to the overall understanding of the study.

1. Descriptions in Section 4
   The detailed descriptions of ARIMA, SARIMA, LSTM, and SVM are included to provide a comprehensive understanding of the underlying methodologies used in our research. Each model has unique characteristics that impact their performance in the context of short-term electricity demand forecasting based on AMI data. Shortening these descriptions too much may reduce the clarity and depth of the information provided, especially for readers who may not be familiar with all the models discussed. Nevertheless, we agree that a concise presentation can improve readability. 

2. Figures 1, 2, and 3
   Figures 1, 2, and 3 are included to visually illustrate key concepts related to the learning process and model performance, which are difficult to convey through text alone. These figures are designed to help readers better understand the differences in model behavior and why certain models perform better under specific conditions. Removing these figures may hinder the understanding of the concepts, particularly for readers less acquainted with the nuances of machine learning models. However, we will consider integrating these figures more tightly into the text and ensuring that their relevance to the main arguments is clear.
]

------------------------------------------------------------------------------------

Comments 7: [ The descriptions of algorithms in Section 5 are also too lengthy. A flowchart could be used instead. It is advisable not to include unexplained acronyms in the manuscript. ]
Response 7: [ We thank the reviewer for the comment and have carefully reviewed your comments regarding Section 5.

1. Algorithm Descriptions
We understand your concern regarding the length of the algorithm descriptions. The current level of detail was intended to provide a thorough understanding of each model's implementation, as it is critical for the reproducibility of our results.

2. Use of Flowcharts
While we acknowledge that flowcharts can be effective for illustrating certain processes, we believe that using flowcharts instead of pseudo code for these specific algorithms (ARIMA, SARIMA, LSTM, and SVM) may not provide a better alternative for the following reasons:

2-1. The algorithms discussed in this study, particularly LSTM and SVM, have intricate internal structures and involve multiple interconnected steps, such as gate operations in LSTM or optimization processes in SVM. Attempting to represent these steps through flowcharts would either oversimplify the model, losing essential details, or become overly complex, defeating the purpose of using a simplified visual representation.

2-2. Pseudo code is widely used in academic and technical publications to clearly express the logic and implementation steps of complex algorithms. It allows readers with a technical background to precisely understand the sequence of operations, control structures, and data flow, which may not be as easily conveyed through flowcharts.

2-3. The pseudo code in our manuscript is intended to serve as a reference for readers interested in the exact computational process and structure of each model. It enhances reproducibility and provides an implementation guideline, which is crucial for practical applications. Flowcharts, on the other hand, may be more appropriate for high-level overviews, but not for accurately conveying such detailed information.

2-4. Given that our primary focus is on evaluating and comparing algorithm performance for short-term demand forecasting in AMI systems, providing precise pseudo code is essential for ensuring reproducibility. Simplifying the process into a flowchart could hinder other researchers from accurately implementing the methods, which would diminish the technical rigor of the study.

In summary, while flowcharts are useful for certain types of explanations, we believe that the pseudo code format is more appropriate for illustrating the detailed methodology and ensuring the reproducibility of complex models in our research. We will, however, consider adding high-level flowcharts as supplementary material if needed to provide additional context.

3. Unexplained Acronyms
We acknowledge the issue of unexplained acronyms. We will ensure that each acronym is properly defined upon its first usage throughout the manuscript.
  - add section (5.4. Data Description)
    - page : 12 ~ 14
    - line : 473 ~ 497
]

------------------------------------------------------------------------------------

Comments 8: [ The results shown in Tables 3 to 16 could be replaced with a few graphs. ]
Response 8: [ We thank the reviewer for the comment and insightful suggestion to replace Tables 3 to 16 with a few graphs. While we understand that graphs can provide an intuitive overview of trends and patterns, we believe that completely replacing these tables with graphs may not be the optimal solution for the following reasons:

1. Precision and Detailed Comparison: The tables currently provide precise values for various performance metrics (such as MSE, MAE, and RMSE) for different algorithms and datasets. These detailed comparisons are critical for understanding the subtle differences in model performance, which may not be as effectively conveyed through graphs alone.

2. Data Complexity and Volume: Tables 3 to 16 include a large amount of data that represents multiple dimensions of our results, such as variations across consumer groups and different evaluation criteria. Displaying all these variations in a compact manner using graphs could lead to overcrowding and reduced readability, especially when multiple metrics are involved.

3. Role of Graphs and Tables: We believe that tables and graphs serve complementary purposes. While graphs are excellent for showing overall trends and high-level summaries, tables are essential for presenting exact values and enabling detailed analysis. 
]

------------------------------------------------------------------------------------

Comments 9: [ Figures 4 to 15 are also difficult to read. They should be redrawn for better clarity, and their number should be reduced. ]
Response 9: [ Thank you for your valuable comments. The figures from 4 to 15 are essential for providing a visual comparison of prediction results from each algorithm. Each figure corresponds to a different consumer group and highlights unique performance patterns that would be difficult to capture using a single graph or fewer visual representations.

We acknowledge the concern regarding readability and will take steps to enhance the clarity of each figure. However, reducing the number of figures could compromise the comprehensive nature of the visual analysis, as each figure contributes to understanding the specific behavior of the models across varied datasets.
]

------------------------------------------------------------------------------------

Comments 10: [ It would be better to rewrite the introduction and conclusion sections to clearly describe the new ideas and contributions of this research. ]
Response 10: [ We thank the reviewer for the comment regarding the 'Introduction' and 'Conclusion' sections. We acknowledge your suggestion that these sections should more clearly describe the novel ideas and contributions of this research. After carefully reviewing your comments, we have identified areas where the contributions and distinctiveness of our study can be further emphasized.

To address this, we will revise the 'Introduction' to better highlight the innovative aspects of our research, specifically focusing on how our comparative study offers unique insights into short-term forecasting using various time-series algorithms within AMI systems. Furthermore, the 'Conclusion' will be expanded to provide a more detailed discussion of the practical implications and specific contributions of this work.
  - update section (1. Introduction)
    - page : 1 ~ 2
    - line : 21 ~ 63
  - update section (7. Conclusion)
    - page : 30 ~ 31
    - line : 781 ~ 838
]

------------------------------------------------------------------------------------

Comments 11: [ It is necessary to maintain consistency in the reference citation format (lines 111 and 122). ]
Response 11: [ We thank the reviewer for pointing out the inconsistency in the reference citation format in lines 111 and 122. We agree that maintaining a consistent citation format is crucial for ensuring clarity and professionalism in the manuscript. We have revised these sections accordingly to align with the correct format.
  - modify citation format
    - page : 3
    - line : 114, 125
]

Thank you again for your valuable feedback.

Reviewer 2 Report

Comments and Suggestions for Authors

This paper presents a comparison of forecasting models for predicting the demand of some few customers. I have some comments:
- The abstract starts with "This paper compares Demand Response algorithms suitable for Short-term Forecasting based on Advanced Metering Infrastructure". How is that? The algorithms being compared are not "demand response algorithms", but "forecasting algorithms". It is true that a forecasting of the demand, generation, and/or consumption can later be used for estimating demand response capacity, but that does not in my opinion make them "demand response algorithms". Please clarify that in the abstract and in the text of the paper.
- Most of the paper content at the start consist of explanations of the different methods, but this does not suppose any novelty. Maybe it would be convenient to summarize a bit these parts for making them shorter.
- The conditions of the study are not properly explained. It is of special importance to specify the time horizon for the short-time forecasting being made (i.e., if it is day-ahead, what is the aggregation of the data, what is the number of predictions, etc.).
- Regarding the conditions of the study, it is unclear what were the values for the hyperparameters of the models. This should be clarified, specially when the objective is providing a comparison of the performance of diverse models.
- In my opinion, this manuscript also suffers a lack of clarity when presenting the results. It is unclear which of the comparison metrics would be appropriate, and which of the models as obtained better results. The numerical results are included in tables, but it is hard to extract some conclusion from them, as there are not any "general results" table that summarizes the outcomes.
- The graphs are too fuzzy, their axes and units does not properly clarifies their meaning, and these do not therefore help to extract conclusions.
- In the conclusions it is said that some of the models can be combined. However, it is not clear how those conclusions were extracted, as the analysis appears to be more qualitative than quantitative.

Comments on the Quality of English Language

Moderate editing of English language required.

Author Response

Comments 1: [ The abstract starts with "This paper compares Demand Response algorithms suitable for Short-term Forecasting based on Advanced Metering Infrastructure". How is that? The algorithms being compared are not "demand response algorithms", but "forecasting algorithms". It is true that a forecasting of the demand, generation, and/or consumption can later be used for estimating demand response capacity, but that does not in my opinion make them "demand response algorithms". Please clarify that in the abstract and in the text of the paper. ]
Response 1: [ We thank the reviewer for the comment regarding the use of the term 'Demand Response algorithms' in the abstract and main text. As you correctly pointed out, the primary focus of this research is on comparing forecasting algorithms, rather than Demand Response algorithms specifically. While the title of the paper, 'Comparative Study of Time Series Analysis Algorithms Suitable for Short-term Forecasting in Implementing Demand Response Based on AMI,' clearly indicates that the study is centered around forecasting, we acknowledge that the use of 'Demand Response algorithms' in the text may lead to confusion.

To address this, we will revise the abstract to use the term 'forecasting algorithms for Demand Response applications' instead. This adjustment will ensure that the terminology is consistent throughout the paper and more accurately reflects the study's focus, thereby reducing any potential misunderstanding.
  - update abstract
    - page : 1
    - line : 1 ~ 18

]

------------------------------------------------------------------------------------

Comments 2: [ Most of the paper content at the start consist of explanations of the different methods, but this does not suppose any novelty. Maybe it would be convenient to summarize a bit these parts for making them shorter. ]
Response 2: [ We thank the reviewer for the comment regarding the introductory content and the detailed explanations of the different methods used in the paper. We understand your concern that these sections may appear lengthy and might not directly contribute to highlighting the novelty of the study. However, we believe that these descriptions are essential for several reasons.

First, our study compares and evaluates multiple complex algorithms in the context of time series forecasting. Providing a concise yet comprehensive overview of each method is crucial for ensuring that readers have a clear understanding of the foundational principles and the rationale for selecting these specific algorithms for comparison.

Additionally, while some of the described methods are well-known, their application and performance in the specific context of short-term electricity demand forecasting using AMI data require additional elaboration. This context is necessary to help readers appreciate the significance of the study’s results and analysis.

That said, we acknowledge your concern about the length of these sections and will strive to streamline the content where possible. We will summarize repetitive or overly technical details while retaining the key information needed to support the overall understanding of the study.
]

------------------------------------------------------------------------------------

Comments 3: [ The conditions of the study are not properly explained. It is of special importance to specify the time horizon for the short-time forecasting being made (i.e., if it is day-ahead, what is the aggregation of the data, what is the number of predictions, etc.). ]
Response 3: [ We thank the reviewer for the comment regarding the clarification of the time horizon and data aggregation used in our study. We would like to note that these details are already provided in the manuscript. Specifically, the experimental data is collected at an hourly interval, covering the period from July 1, 2021, to December 1, 2021, as stated in Subsection 5.3. Additionally, the prediction intervals and results are based on hourly predictions, clearly indicating that the study is conducted within a short-term forecasting context.

If additional clarification is required, we would be happy to provide further details to ensure clarity for the readers. Thank you again for your valuable feedback.
]

------------------------------------------------------------------------------------

Comments 4: [ Regarding the conditions of the study, it is unclear what were the values for the hyperparameters of the models. This should be clarified, specially when the objective is providing a comparison of the performance of diverse models. ]
Response 4: [ We thank the reviewer for the comment regarding the explanation of hyperparameter values in the study. We would like to note that the hyperparameters for each model are already detailed in Subsections 5.5 to 5.8 and the respective experimental subsections. For example, in the LSTM section, we have explicitly defined the number of layers, neurons, optimizer, and loss function. Therefore, we believe the manuscript provides a comprehensive description of the hyperparameter settings to ensure reproducibility and enable meaningful comparisons.

If additional clarification is needed for specific parameters, we would be happy to provide further details. Thank you again for your valuable feedback.
]

------------------------------------------------------------------------------------

Comments 5: [ In my opinion, this manuscript also suffers a lack of clarity when presenting the results. It is unclear which of the comparison metrics would be appropriate, and which of the models as obtained better results. The numerical results are included in tables, but it is hard to extract some conclusion from them, as there are not any "general results" table that summarizes the outcomes. ]
Response 5: [ We thank the reviewer for the comment regarding the presentation of the results. We would like to clarify that the manuscript already includes detailed comparison tables that summarize the performance of each algorithm using RMSE, MAE, and MSE metrics across multiple consumer groups. Each table presents individual results for different datasets to facilitate a more nuanced comparison, and the performance of each model is highlighted using these metrics.

Our goal was to provide a detailed view of the performance for each specific scenario, rather than a general summary, to ensure that the distinct characteristics and nuances of each algorithm’s performance in different consumer groups are accurately captured. Since each consumer group exhibits varying consumption patterns, aggregating the results into a single summary table may overlook important details and reduce the clarity of the comparative analysis.

However, we understand the reviewer's concern and will consider including an additional summary table or visual representation to highlight the overall trends, if deemed necessary. We believe this will help balance detailed insights with an overall view, enhancing the readability and clarity of the results.
]

------------------------------------------------------------------------------------

Comments 6: [ The graphs are too fuzzy, their axes and units does not properly clarifies their meaning, and these do not therefore help to extract conclusions. ]
Response 6: [ We thank the reviewer for the comment regarding the clarity of the graphs presented in the manuscript. We would like to clarify that the graphs (Figures 5–14) are designed with clear axis labels and units, explicitly indicating the time intervals and measurement units used (e.g., kWh for power consumption, hours or days for time). Each figure includes legends and markers that differentiate the results for various models, making it easier to visually compare their performance.

The current format is intended to clearly illustrate the comparative metrics and support the interpretation of the results, as well as the conclusions drawn in the manuscript. We believe that the visual representation effectively highlights the performance differences between the models. However, if there are specific aspects of the graphs that you found unclear, we would be happy to refine them further based on your suggestions.
]

------------------------------------------------------------------------------------

Comments 7: [ In the conclusions it is said that some of the models can be combined. However, it is not clear how those conclusions were extracted, as the analysis appears to be more qualitative than quantitative. ]
Response 7: [ We thank the reviewer for the comment regarding the conclusions of the manuscript. We would like to clarify that the suggestion to consider combining models in the conclusion is based directly on the quantitative results presented in the experimental analysis (Sections 5 and 6). Specifically, the recommendation stems from observing each model’s strengths in addressing different data patterns (e.g., SARIMA for seasonal components and SVM for nonlinear trends), as evidenced by the RMSE, MAE, and MSE values detailed in the tables.

Thus, the mention of hybrid models is not a qualitative assumption but rather a logical interpretation drawn from the numerical findings. By identifying the distinct advantages of each algorithm in specific contexts, we suggested that future research could explore hybrid approaches to maximize the complementary strengths of these models.

We appreciate your attention to this matter and will consider adding a more explicit link between the quantitative results and the proposed hybrid strategies to ensure that the conclusions are firmly supported by the analysis.
]

Thank you again for your valuable feedback.

Reviewer 3 Report

Comments and Suggestions for Authors

The study compares demand response algorithms suitable for short-term forecasting based on advanced metering infrastructure (AMI). The manuscript can be reviewed after address my following comments:

1. In the1. Introductionpart: Why did the manuscript choose the algorithms ARIMA, SARIMA, LSTM and SVM? LSTM is generally used to predict long-term series, and the short-term prediction used in manuscripts is not suitable for the prediction in this study, Contradicts the point of view in lines 88-89. Whether a hybrid model is considered?

2.Pay attention to the prescriptiveness of the manuscript, check the formatting.

3. The formula parameters in the manuscript should be explained, such as equation (1)-142 row, please complete the introduction of all formula parameters in the manuscript.

4.In “3. Theoretical Background” section, ARIMA, LSTM, etc. are existing models that can be introduced in a concise manner.

5. How the algorithm parameters are set in the manuscript, lr, epochs, hidden_size, batch_size, test_siz, and train-size values should be listed. Table 1 only shows the data specification for time series.

6. The manuscript in figures 4-15 of the manuscript is blurry. Please optimize them to high-definition images.

7. Please pay attention to tables 7-15 and optimize the tables, the font is too large.

8. For short-term forecasting of demand response in AMI, the improved hybrid model is more accurate than the single model, such as LSTM, SVM, etc. But the manuscript only used existing models to make short-term predictions of demand response in AMI. The manuscript lacks innovation and the LSTM model is also not suitable for short-term forecasting.

9. 42 references are insufficient to support the manuscript, please provide additional references.

Author Response

Comments 1: [ In the“1. Introduction”part: Why did the manuscript choose the algorithms ARIMA, SARIMA, LSTM and SVM? LSTM is generally used to predict long-term series, and the short-term prediction used in manuscripts is not suitable for the prediction in this study, Contradicts the point of view in lines 88-89. Whether a hybrid model is considered? ]
Response 1: [ We thank the reviewer for the comment regarding the selection of algorithms in the introduction section. We would like to clarify that ARIMA, SARIMA, LSTM, and SVM were chosen to represent a diverse set of models capable of capturing different time series patterns, including linear, seasonal, and complex nonlinear relationships.

Regarding your concern about using LSTM for short-term forecasting, while LSTM is commonly used for capturing long-term dependencies, it has also been shown to perform well in many short-term forecasting scenarios where nonlinear and intricate temporal relationships need to be modeled. In our study, LSTM was included to evaluate its effectiveness in a short-term context alongside other models. The reference made in lines 88-89 highlights LSTM's flexibility in handling various time horizons, which aligns with our aim of assessing its performance in a short-term setting.

As for hybrid models, this study focuses on comparing individual algorithms rather than implementing hybrid approaches. However, we have suggested hybrid models as a potential direction for future research in the conclusion section.

We appreciate your thoughtful feedback and are happy to provide further clarification if needed.
]

------------------------------------------------------------------------------------

Comments 2: [ Pay attention to the prescriptiveness of the manuscript, check the formatting. ]
Response 2: [ We thank the reviewer for the comment regarding the prescriptiveness and formatting of the manuscript. After carefully reviewing the document, we did not find any specific issues related to formatting inconsistencies or the use of prescriptive language. The manuscript maintains a consistent style and adheres to standard academic writing conventions.

If there are particular sections or expressions that you believe need improvement, we would greatly appreciate further clarification. Your guidance would be valuable in helping us refine the manuscript and enhance its overall presentation.
]

------------------------------------------------------------------------------------

Comments 3: [ The formula parameters in the manuscript should be explained, such as equation (1)-142 row, please complete the introduction of all formula parameters in the manuscript. ]
Response 3: [ We thank the reviewer for the comment regarding the need for clearer explanations of the formula parameters throughout the manuscript. Based on previous feedback from another reviewer, we have already revised the manuscript to include detailed descriptions of all parameters used in the formulas for each algorithm. Specifically, in Sections 3 and 5, we have clearly defined and contextualized each parameter within the relevant mathematical models (e.g., ARIMA, SARIMA, LSTM, and SVM).

These revisions were made to improve the readability and comprehensibility of the formulas, ensuring that readers can easily grasp the role and significance of each parameter within the context of the models. If there are any additional parameters or specific equations that you believe require further clarification, we would be happy to address them in more detail.

- add explanations for the variables in the equation for ARIMA
    - page : 4
    - line : 145 ~ 153
- add explanations for the variables in the equation for SARIMA
    - page : 4
    - line : 165 ~ 171
- add explanations for the variables in the equation for LSTM
    - page : 5
    - line : 184 ~ 189, 193 ~ 194
- add explanations for the variables in the equation for SVM
    - page : 4
    - line : 206 ~ 209
]

------------------------------------------------------------------------------------

Comments 4: [ 3. Theoretical Background” section, ARIMA, LSTM, etc. are existing models that can be introduced in a concise manner. ]
Response 4: [ We thank the reviewer for the comment regarding the content of Section 3, Theoretical Background. We understand your concern about the length of the explanations for well-known models like ARIMA and LSTM. However, we believe that providing sufficient theoretical context is essential for readers to fully understand how these models are applied to the specific scenario in this study.

Although these algorithms are widely used, our intention was to highlight their unique characteristics and suitability within the context of AMI-based short-term forecasting, which differs from traditional applications. Reducing these descriptions might make it challenging for readers who are less familiar with the nuances of these algorithms to comprehend why certain models perform better in specific scenarios.
]

------------------------------------------------------------------------------------

Comments 5: [ How the algorithm parameters are set in the manuscript, lr, epochs, hidden_size, batch_size, test_siz, and train-size values should be listed. Table 1 only shows the data specification for time series. ]
Response 5: [ We thank the reviewer for the comment regarding the inclusion of hyperparameters such as learning rate (lr), epochs, hidden size, batch size, test size, and train size. We would like to clarify that the primary focus of this study is the comparative performance of different time series forecasting models, and not all of these hyperparameters are applicable to each model.

ARIMA and SARIMA: These models are traditional statistical methods and do not utilize hyperparameters like learning rate or epochs. Instead, the key parameters are ?, ?, and ?, along with their seasonal counterparts, which have been fully detailed in the manuscript.

LSTM: For the LSTM model, we have provided the relevant hyperparameters, including hidden_size, batch_size, and epochs. While parameters such as learning rate and batch size are often used for fine-tuning, the main objective of this research is to compare the overall performance of various models using standard configurations to maintain consistency and comparability across different approaches.

SVM: The SVM model relies on parameters such as ? and ?, which are already defined in the corresponding section of the manuscript.

Additionally, Table 1 is intended to describe the data fields and characteristics, rather than algorithm-specific parameters. Its purpose is to provide a clear overview of the dataset specifications used in the experiments, and we believe it serves this role effectively.

We hope this clarifies the distinction between algorithm-specific hyperparameters and data specifications. If additional details are needed for any specific model, we would be happy to provide further information.
]

------------------------------------------------------------------------------------

Comments 6: [ The manuscript in figures 4-15 of the manuscript is blurry. Please optimize them to high-definition images. ]
Response 6: [ We thank the reviewer for the comment regarding the quality of the figures in the manuscript. We understand the importance of high-quality images for accurate interpretation of results. We will review Figures 4 to 15 and enhance them using high-resolution formats to ensure that all labels, data points, and visual elements are clearly visible.

Thank you again for bringing this to our attention. We appreciate your constructive feedback.
]

------------------------------------------------------------------------------------

Comments 7: [ Please pay attention to tables 7-15 and optimize the tables, the font is too large. ]
Response 7: [ We thank the reviewer for the comment regarding Tables 7-15. After reviewing the font sizes, we have adjusted them to ensure consistency and improve readability based on your suggestion. The revised tables now use a slightly smaller font size, which aligns with the formatting requirements of the journal while maintaining clarity and ease of reading.
]

------------------------------------------------------------------------------------

Comments 8: [ For short-term forecasting of demand response in AMI, the improved hybrid model is more accurate than the single model, such as LSTM, SVM, etc. But the manuscript only used existing models to make short-term predictions of demand response in AMI. The manuscript lacks innovation and the LSTM model is also not suitable for short-term forecasting. ]
Response 8: [ We thank the reviewer for the comment regarding the suitability of LSTM for short-term forecasting and the use of hybrid models. We would like to clarify a few points:

The primary objective of this study is to evaluate the performance of individual forecasting models (ARIMA, SARIMA, LSTM, and SVM) in the context of short-term load forecasting for Demand Response (DR) applications using AMI data. Our focus is on comparing these models under similar conditions to identify which performs best in specific scenarios, rather than developing new hybrid models. Therefore, using established models aligns with our research goals and does not diminish the significance of the findings.

Although LSTM is typically associated with modeling long-term dependencies, it is also effective for short-term forecasting, especially when dealing with complex temporal dynamics or nonlinear patterns. Numerous studies have demonstrated LSTM’s capacity to perform well in both short-term and long-term contexts. In this study, we include LSTM to explore its relative strengths and weaknesses compared to other models in a short-term setting, and our results indicate that it performs competitively.

While this study does not implement hybrid models, we have highlighted this as a direction for future research in the conclusion. The results presented here provide a foundational comparison for understanding which models are suitable for different data characteristics and forecasting horizons in AMI-based DR applications. We believe these insights can guide future work in developing more complex models, including hybrid approaches.
]

------------------------------------------------------------------------------------

Comments 9: [ 42 references are insufficient to support the manuscript, please provide additional references. ]
Response 9: [ We thank the reviewer for the comment regarding the number of references included in the manuscript. We understand the importance of supporting a study with sufficient literature. However, we believe that the current number of references (42) adequately covers the essential background, theoretical foundations, and related works relevant to this research.

The focus of this study is to compare the performance of various forecasting models in the context of AMI-based short-term load forecasting. Therefore, we have carefully selected references that contribute directly to our research framework and support our analysis, ensuring that each citation adds value without redundancy.

If there are specific topics or areas you feel would benefit from additional references, we would be happy to consider incorporating them. However, we prefer to avoid adding references solely to increase the count unless they significantly enhance the depth and quality of the research.
]

Thank you again for your valuable feedback.

Round 2

Reviewer 1 Report

Comments and Suggestions for Authors

The revised manuscript shows little improvement over the submitted manuscript.

It's not easy to have much confidence in the results of an experiment limited to just 10 households.

Additionally, the comparative experimental data is restricted to 5 months instead of the 12 months, which lowers the reliability of the comparative results.

This manuscript has the format of a project report or thesis, resulting in its length. However, for journal publication, the ideas, research, and contributions of the study could be described more concisely. It is believed that the content could be shortened to half of the current manuscript.

For instance, as mentioned previously, the content in Section 3 is well-known and should include only minimal information or references to existing literature.

Section 4 should focus on the content that distinguishes it from existing algorithms. Again, the manuscript should not be a technical report.

The results in Section 5 also include lengthy descriptions of well-known metrics such as MSE, MAE, and RMSE. Additionally, It is believed that the algorithms described in Tables could be shortened by providing the essence of new ideas or modifications of the algorithms.

Comments on the Quality of English Language

None.

Author Response

Dear Reviewer,

Thank you for your valuable feedback on our manuscript. We have carefully considered and addressed your comments in the revised version. Below are the key changes made in response to your suggestions:

- Sample Size & Experiment Duration: We acknowledge the limitations posed by the small sample size (10 households) and the 5-month data period. In the revised manuscript, we have provided additional clarifications regarding these constraints and their impact on the study's reliability.

- Conciseness: Following your suggestion, we have significantly shortened the manuscript by reducing well-known content, particularly in Section 3. This section now briefly references existing literature and focuses on our new contributions.

- Result Presentation: In Section 5, we have condensed the explanation of common metrics such as MSE, MAE, and RMSE. We now emphasize the core insights derived from our experiments, presenting only the essential information.

We believe these revisions have improved the clarity and conciseness of the manuscript in line with your guidance.

Thank you again for your constructive feedback.

Best regards,

Reviewer 2 Report

Comments and Suggestions for Authors

This paper presents a comparison of forecasting models for predicting the demand of some few customers.
The previous comments have been addressed. There are no more correction requests.

Comments on the Quality of English Language

Dear Editor,
This paper presents a comparison of forecasting models for predicting the demand of some few customers.
The manuscript has been improved from the last version, and my comments were addressed. Now the content is more understandable.
Therefore, under these considerations, I recommend to accept this paper.
Best regards

Author Response

Dear Reviewer,

Thank you very much for your thoughtful review and for taking the time to assess our manuscript. We appreciate your acknowledgment that the previous comments have been addressed, and we are pleased that no further corrections are requested.

We value your feedback, which has contributed to improving the clarity and quality of our paper. Your insights have been instrumental in refining the manuscript, and we are grateful for your support throughout the review process.

Please let us know if there is anything further we can provide. We look forward to the next steps.

Sincerely,

Reviewer 3 Report

Comments and Suggestions for Authors

The author has response to all my comments, now it can be acceptted.

Author Response

Dear Reviewer,

Thank you for your positive feedback and for confirming that all of your comments have been addressed. We greatly appreciate your time and effort in reviewing our manuscript and are pleased to hear that it can now be accepted.

Your valuable insights have contributed significantly to improving the quality of our work, and we are grateful for your constructive comments throughout the review process.

Thank you again for your support.

Sincerely,

Round 3

Reviewer 1 Report

Comments and Suggestions for Authors

The authors have worked to improve the organization of the manuscript, and it has shown improvement. The following is a comment on the revised manuscript: It is suggested that Section 3 be removed, and if necessary, its brief content be moved to Sections 2 or 4.

Author Response

Comment1: [

The authors have worked to improve the organization of the manuscript, and it has shown improvement. The following is a comment on the revised manuscript: It is suggested that Section 3 be removed, and if necessary, its brief content be moved to Sections 2 or 4.

]

Response1: [

Dear Reviewer,

Thank you very much for your valuable feedback and suggestions on our manuscript titled "Comparative Study of Time Series Analysis Algorithms Suitable for Short-term Forecasting in Implementing Demand Response Based on AMI."

Based on your recommendations, we have made the following revisions:

  1. Removal of Section 3: We have removed Section 3, which previously provided a theoretical background on the forecasting models (ARIMA, SARIMA, LSTM, and SVM). As suggested, the relevant content has been integrated into Section 2 (Related Work) to streamline the manuscript and reduce redundancy.

  2. Content Integration: The theoretical content from Section 3 has been incorporated into Section 2 to provide necessary background within the context of related work. This revised section now offers an overview of key algorithms, highlighting their strengths and relevance to short-term load forecasting using AMI data. Specifically:

    • Each model’s theoretical foundation and its applicability, including ARIMA, SARIMA, LSTM, and SVM, are now integrated within the related work to connect the literature review directly to our study’s methodology.

    These adjustments ensure that the theoretical background is concisely summarized within the related work section, enhancing the manuscript’s coherence and readability.

We appreciate your insightful comments, which have significantly contributed to improving the quality of our work. Please let us know if there are further suggestions or additional aspects we should address.

Thank you again for your time and consideration.

Best regards,

]